# CiFi: accurate long-read chromosome conformation capture with low-input requirements

Sean P. McGinty[1,5], Gulhan Kaya [1,5], Sheina B. Sim [2], Alex Makunin [3], Renée L. Corpuz [2], Michael A. Quail [3], Mohamed Abuelanin [1], Mara K. N. Lawniczak [3], Scott M. Geib [2], Jonas Korlach [4] ✉ & Megan Y. Dennis [1] ✉

Hi-C characterizes three-dimensional chromatin organization, facilitates haplotype phasing, and enables genome-assembly scaffolding, but encounters difficulties across complex regions. By coupling chromosome conformation capture (3C) with PacBio HiFi long-read sequencing, here we develop a method (CiFi) that enables analysis of genomic interactions across repetitive regions. Starting with as little as 60,000 cells (sub-microgram DNA), the method produces multi-kilobasepair HiFi reads that contain multiple interacting, concatenated segments (~350 bp to 2 kbp). This multiplicity and increase in segment length versus standard short-read-based Hi-C improves read-mapping efficiency and coverage in repetitive regions and enhances haplotype phasing. CiFi pairwise interactions are largely concordant with Hi-C from a human lymphoblastoid cell line, with gains in assigning topologically associating domains across centromeres, segmental duplications, and human disease-associated genomic hotspots. As CiFi requires less input versus established methods, we apply the approach to characterize single small insects: assaying chromatin interactions across the genome from an *Anopheles coluzzii* mosquito and producing a chromosome-scale scaffolded assembly from a *Ceratitis capitata* Mediterranean fruit fly. Together, CiFi enables assessment of chromosome-scale interactions of previously recalcitrant low-complexity loci, low-input samples, and small organisms.

Hi-C, a sequencing technique that assays chromosome conformation capture (3C) genome-wide[1], provides insights into chromatin organization and *cis*-regulatory interactions. The method relies on in situ proximity ligations produced by endonuclease cutting of cross-linked chromatin in cells, that results in preferential binding of genomic fragments positioned physically near to each other in the nucleus. Several layers of genome folding create higher-order genome organization spanning from the largest level of topologically associating domains (TADs) (~1 Mbp), down to the intermediate level of TAD subdomains (100–500 kbp), to the smallest level of regulatory loops (1–100 kbp)[2]. The regulation of gene expression involves the activities and relationships between multiple *cis*-regulatory elements, including the promoter (typically <1 kbp from the transcription start site) and enhancers that can act over longer distances (sometimes

[1]Genome Center, MIND Institute, and Department of Biochemistry & Molecular Medicine, University of California, Davis, Davis, CA, USA. [2]U.S. Department of Agriculture, Agricultural Research Service, Tropical Pest Genetics and Molecular Biology Research Unit, U.S. Pacific Basin Agricultural Research Center, Hilo, HI, USA. [3]Wellcome Sanger Institute, Hinxton, UK. [4]Pacific Biosciences, Menlo Park, CA, USA. [5]These authors contributed equally: Sean P. McGinty, Gulhan Kaya. ✉e-mail: jkorlach@pacb.com; mydennis@ucdavis.edu

**Fig. 1 | CiFi produces unbiased 3C concatemer long reads. A** Overview of the CiFi approach. Created by BioRender. Dennis, M. (2025) https://BioRender.com/2d02ebg. **B** Normalized read coverage comparison of Sequel II data for DpnII 3C libraries generated without (Standard) and with the amplification-protocol (CiFi) for human chromosome 1 of GM12878 (*excluding non-unique regions including the centromere; cytogenetic bands (gray, white, black) and the centromere regions (striped and criss-cross pattern) are depicted on the chromosome ideogram. Normalized read coverages of additional autosomal chromosomes are depicted in Supplementary Fig. 1. **C** Segment-number distribution per HiFi read and **D** length distribution of concatemer segments for DpnII and HindIII CiFi libraries, with medians indicated. Source data are provided as a Source Data file.

more than a megabasepair)[3,4]; thus, interruptions to chromatin interactions through genetic variants can have significant functional consequences contributing to diverse phenotypes and diseases.

Given that a majority of reads represent intrachromosomal *cis* interactions, Hi-C is also an important tool for haplotype phasing and genome assembly[5,6]. For example, when one Hi-C read can be phased, it is possible to impute the same haplotype to a paired read mapping to the same chromosome; this is effectively used to bin reads into different alleles, facilitating phased assemblies. Further, it enables scaffolding when read pairs span assembled contigs. All sequencing data used for genome assembly, phasing, and scaffolding ideally derive from the same individual to reduce bioinformatic complexity, but this can present a challenge when working with small organisms with very little starting material available to produce multiple sequence library preparations with large input requirements.

Short-read sequencing is classically employed in Hi-C with paired-end reads representing two interacting regions. While effective across much of the genome, short reads can fail to map uniquely across repetitive and low-complexity regions. Further, higher-order chromatin interactions that exist within the nucleus are difficult to parse when looking through a pairwise lens. Multi-contact 3C (MC-3C)[7] and Pore-C[8] have recently been developed that combine 3C with long-read sequencing (PacBio and nanopore, respectively), enabling a more complete picture of proximally-interacting concatemers. While these approaches are effective in disentangling three-dimensional (3D) genome folding at the allele level[9,10], both require significant starting material (e.g., 10 million cells for Pore-C), that have made them inaccessible to certain applications and organisms.

To overcome this challenge, here we develop CiFi that couples 3C with PacBio HiFi sequencing that leverages unbiased whole-genome amplification[11] to produce high-quality long-read mappings and chromatin interactions from low starting materials. Using this approach, we successfully produce multicontact CiFi reads containing multiple segments of several kilobasepair lengths, enabling improvements in mapping across complex repetitive sequences and expanding the ability to phase haplotypes. Applying this method to human cells and single insects, we demonstrate improvements in resolving chromatin

interactions and produce a chromosome-scale assembly that requires only tens of thousands of cells.

## Results and discussion
### 3C with HiFi sequencing
To develop 3C coupled with PacBio HiFi sequencing, we initially tested library preparations on the human GM12878 lymphoblastoid cell line (LCL) with DpnII restriction endonuclease (as described for Pore-C[8]), but observed low sequence yields (10–20% of standard Sequel II runs; Supplementary Data 1). While SMRT Cell loading was as expected, polymerase read lengths were short (mean 33.4 kbp) and showed a low abundance of HiFi reads (9.8% of polymerase reads; 3.46 Gbp at mean read length 7.24 kbp; Supplementary Data 1). Repeating the experiment on LCLs using the MC-3C protocol yielded a similarly low HiFi yield, which is in line with published work[7,10] (Supplementary Data 1 and Supplementary Table 1). Hypothesizing that residual cross-links remained on the DNA that prevent productive sequencing, following 3C, we implemented a genome-wide amplification-based protocol designed for challenging samples[11], using a high-fidelity PCR enzyme to enrich for uncross-linked molecules ahead of sequencing (Fig. 1A). This dramatically increased raw sequence yields and read lengths to standard sequencing performance (mean 105 kbp) and conversion to HiFi data (49.3% of polymerase reads; 30.6 Gbp at mean read length 9.35 kbp and median read quality value (QV) 38; Supplementary Data 1). The published study[11] amplified genomic DNA from several organisms and observed limited PCR biases; this was also the case for the GM12878 DpnII 3C library, with 1.8% of data representing PCR duplicates and no obvious dropouts evident when comparing read coverage with and without amplification (Fig. 1B and Supplementary Fig. 1).

To investigate the effects of multi-contact segment lengths and their resulting resolution of repetitive genomic regions, we performed 3C with DpnII (4 cutter) and HindIII (6 cutter), respectively, followed by amplification, size selection (>5 kbp), and HiFi sequencing (Supplementary Fig. 2). From the resulting HiFi reads (median lengths of 7.6 kbp; Supplementary Data 1), subsequent in silico deconcatenation produced a median of 17 segments at 350 bp for DpnII and 2 segments at 1893 bp for HindIII (Fig. 1C, D and Supplementary Data 2). The use of

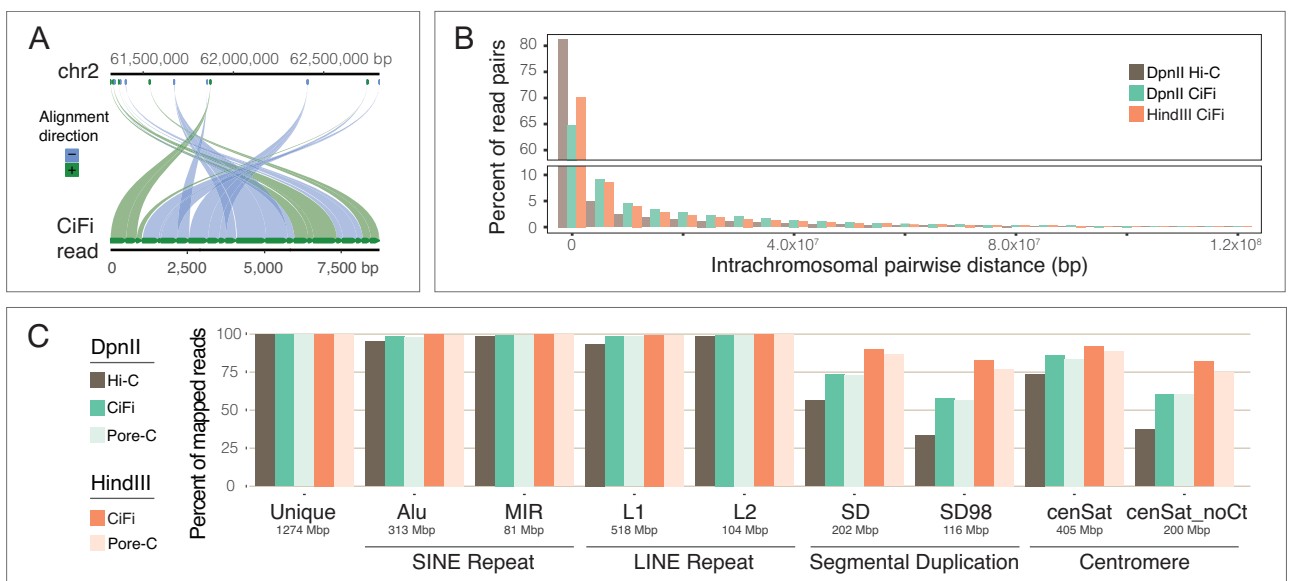

**Fig. 2 | Comparisons of read mappings between short- and long-read 3C approaches. A** An example of mapping positions of individual segments across human chromosome 2 from a single DpnII CiFi read. **B** Histogram of pairwise intrachromosomal distances between DpnII and HindIII CiFi segments from the same HiFi read and DpnII Hi-C paired reads plotted as a percent of total read pairs mapping intrachromosomally. **C** Percent of all mapped reads with MAPQ score of one or higher for Hi-C with Illumina[13] versus 3C with CiFi and Pore-C[8] across different repetitive genome classifications (short interspersed nuclear elements (SINEs) *Alu* and *Mir*, long interspersed nuclear elements (LINEs) *L1* and *L2*, segmental duplications at both 90% (SD) and 98% (SD98) identity, and centromeres with and without the centromeric transition (CT) regions) with sizes of each type indicated. Percent of mapped reads and coverage at varied MAPQ score cutoffs can be found in Supplementary Figs. 4 and 5, respectively. Source data are provided as a Source Data file.

different restriction enzymes enables flexible resolution control, with DpnII providing more interactions and data from a single CiFi read, but with shorter segment lengths, while HindIII produces less pairwise interactions but with longer segment lengths.

## Improvements in read mapping and phasing

We next mapped CiFi concatemer reads against the human reference genome (T2T-CHM13v2.0[12]) and found that a majority of the segments map to homologous chromosomes for both DpnII and HindIII, a pattern consistent across both unique (80.0%) and repetitive (79.3%) regions Fig. 2A and Supplementary Fig. 3). Next, we converted segments per HiFi read into paired interactions such that, for example, a single DpnII CiFi read with 17 segments represents 136 pairwise interactions (equivalent to 272 Illumina paired-end reads). Applying this to each CiFi dataset resulted in 1.9 billion interactions for DpnII and 38.5 million interactions for HindIII. Focusing on intrachromosomal pairwise interactions shows the expected 3C decay with increasing distance and spanning all length scales, going as far as >100 Mbp (the average size of a human chromosome; Fig. 2B). Further, both DpnII and HindIII CiFi show proportionally more long-distance intrachromosomal interactions compared with a 101-bp paired-end Illumina Hi-C dataset generated as a standard resource for GM12878[13].

Comparing read mapping metrics between the CiFi and Illumina Hi-C shows better representation and coverage across non-unique genome space, including short interspersed nuclear elements (SINEs) *Alu* and *Mir*, long interspersed nuclear elements (LINEs) *L1* and *L2*, segmental duplications at both 90% (SD) and 98% (SD98) identity, and centromeres with and without the centromeric transition regions (Fig. 2C and Supplementary Figs. 4 and 5). Improvements were most evident across SDs and centromeres, with only 33–37% of Illumina Hi-C reads exhibiting a MAPQ cutoff ≥1, compared with 83–89% of CiFi with HindIII segments. Comparisons with published long-read Pore-C[8] datasets for both DpnII and HindIII show equivalent mapping metrics with CiFi in percentage of reads mapped and coverage (Fig. 2C and Supplementary Figs. 4 and 5).

To assess if the longer reads produced by CiFi improve haplotype phasing, we used a GM12878 telomere-to-telomere assembly[14,15] to assign phase to the mapped reads[16]; this resulted in 23.9% CiFi phased segments versus 6.9% of phased Illumina Hi-C reads. Previous work has shown that ~98% of Hi-C read pairs mapping to homologous chromosomes represent *cis* interactions, with this proportion increasing to over 99.4% for pairs within 30 Mbp[5]. This high rate of *cis* interactions enables haplotype assignment by linking phased reads with their unphased paired reads. We applied a similar principle to CiFi data and found that 97.8% of reads mapping to the same chromosome share the same haplotype when comparing phased segments within individual CiFi reads. Building on this observation, we imputed phase information to homologous segments within CiFi reads that mapped within 30 Mbp of segments with concordantly assigned phases (see "Methods"). This approach enables us to conservatively assign haplotype phase to 80.3% of CiFi segments compared with 10.9% of Hi-C reads (in line with published results for the Illumina dataset[13]). Cumulatively, improved phasing has implications in characterizing chromatin structure using diploid (or polyploid) genomes and producing high-quality phased genome assemblies.

## Resolving chromatin interactions across duplications

Using pairwise data to generate genome-wide contact matrices (Fig. 3A and Supplementary Fig. 6), we found interactions for DpnII CiFi and Illumina Hi-C to correlate broadly across chromosomes at 2.5-Mbp resolution ($r^2 = 0.89$), with high correlation consistent across increasing resolutions (Supplementary Table 2). Unique regions were more highly correlated ($r^2 = 0.92$) versus SDs and centromeres ($r^2 = 0.71$) due to the reduced read coverage of Hi-C (Fig. 3B and Supplementary Fig. 7). We next used chromatin interactions to partition the genome into TADs[17], or regions that interact with each other in 3D chromatin more often than adjacent regions[18]. Applying the measure of concordance (MoC) metric[19]—which assesses basepair overlap between domains considering overall chromosome size—finds high concordance between CiFi and Hi-C

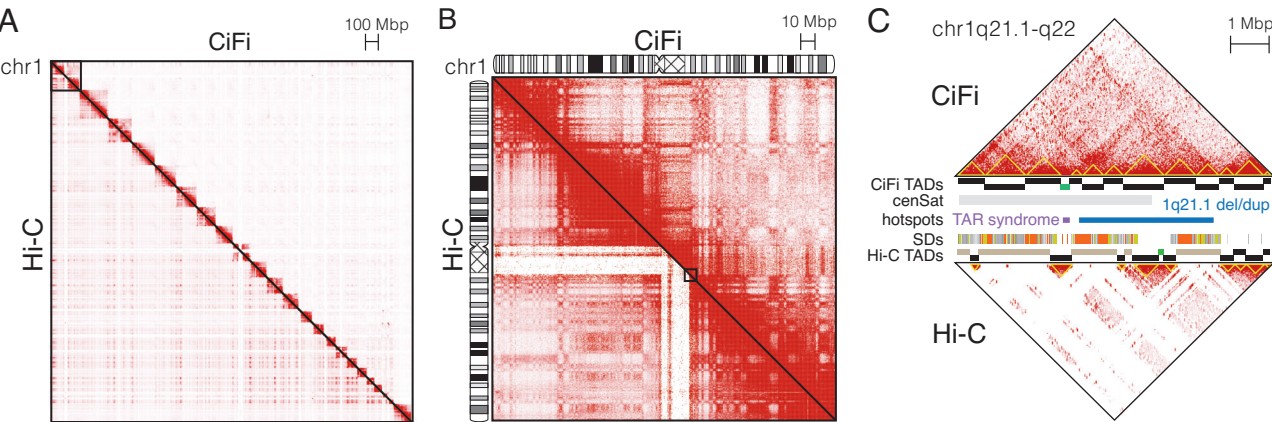

**Fig. 3 | Pairwise chromatin contacts for human LCL GM12878.** DpnII libraries for CiFi (above diagonal) and Hi-C (below diagonal) were used to generate pairwise interaction maps (**A**) genome wide (not normalized and at 2.5-Mbp resolution); and **B** across chromosome 1 (normalized using the Knight–Ruiz algorithm[61] and at 250-kbp resolution). Cytogenetic bands and the centromere (criss-cross pattern) are depicted on each chromosome 1. Increasing red shading corresponds to greater paired interactions. Equivalent plots of CiFi and Pore-C can be found in Supplementary Fig. 9. **C** UCSC Genome Browser view of chr1q21.1–q22 (chr1:142.3 Mbp–150.4 Mbp; T2T-CHM13v2.0) with TopDom[17] topologically associating domain (TAD) classifications ("domain" in black, "boundary" in green, "gap" in tan), centromere satellites (cenSat), and segmental duplications (SDs) track depicted. Two genomic hotspots are delineated, including the disease-associated TAR syndrome locus (purple) and chromosome 1q21.1 deletion (del) and duplication (dup) region (blue). TADs are also depicted as yellow triangles along the contact matrix. Pairwise interactions for CiFi are depicted on the top and Hi-C on the bottom at 10-kbp resolution. Source data are provided through NCBI BioProject accession PRJEB83708.

TADs genome wide (MoC = 0.83) (Supplementary Tables 3 and 4 and Fig. 8). Comparisons with DpnII Pore-C and CiFi show even stronger correlation of contact matrices genome wide at 2.5-Mbp resolution ($r^2 = 0.99$; Supplementary Tables 2–4 and Supplementary Figs. 9 and 10) as well as concordance of identified TADs (MoC = 0.79). Together, CiFi, Pore-C, and Hi-C similarly represent genome organization across a majority of the human genome, with long-read approaches providing notable improvements across repetitive and complex regions.

To better understand the extent to which CiFi improves assessment of chromatin interactions, we cataloged genomic regions classified as gaps in the TAD analysis—representing loci with insufficient data to call domains. Gaps are found less frequently in TADs delineated using CiFi (~2% of human autosome basepairs) compared with Illumina Hi-C (~5%), with more marked differences across repetitive regions (SDs: 8.6% CiFi vs 18% Hi-C; centromeres: 15% CiFi vs 34% Hi-C). These gaps present challenges in analysis of human genomic hotspots—regions enriched for SDs that carry some of the greatest amounts of structural polymorphism in the genome[20]—where we observed none across flanking SDs using CiFi, in contrast to 8% using Hi-C.

One such locus is present on chromosome 1q21.1, where recurrent copy-number variants are associated with human developmental conditions[21,22], including intellectual disability, autism, congenital abnormalities, and thrombocytopenia absent radius (TAR) syndrome (Fig. 3C). While Hi-C TAD analysis of this locus contains mostly gaps due to insufficient interaction data, CiFi TAD analysis shows no gaps. Even in regions where Hi-C TAD domains are identified, discrepancies exist, e.g., across the ~200-kbp TAR syndrome locus, Hi-C classifies the unique portion of the locus as a domain flanked by two gaps spanning the adjacent SDs mediating the disease-associated microdeletion. In contrast, CiFi assigns the locus as a TAD boundary, a region that serves as a barrier between adjacent TADs. While further work is necessary to validate this finding, boundaries have been shown to be enriched for active chromatin and highly expressed genes[23,24]. Several housekeeping genes reside at this locus (*POLR3GL*, *POLR3C*, *RBM8A*), thereby supporting the CiFi result. The use of CiFi to characterize chromatin interactions at genomic hotspots such as this facilitates

understanding of how structural variants impact gene regulation and underlying disease etiologies.

### Scaling down CiFi for smaller sample inputs

To explore the benefits of lower input requirements, we successfully scaled down GM12878 DpnII CiFi samples by over 100-fold, from 10 million cells (~60 μg of DNA input) to 62,000 cells (~370 ng), resulting in consistent sequencing metrics and contact matrices across all starting amounts (Supplementary Data 1, Supplementary Table 5, and Supplementary Fig. 11). Moving beyond human cell lines, we next applied DpnII CiFi to a single male *Anopheles coluzzii* mosquito (~250 ng starting input[25]) and generated 2.37 million HiFi reads and 21.1 million segments of median length 509 bp (Supplementary Data 1 and 2). We mapped these data across the 263-Mbp reference genome (AcolN3) and compared with Hi-C reads from a female *An. coluzzii* generated with two four-cutter restriction enzymes used to aid in the assembly of AcolN3 (NCBI RefSeq GCF_943734685.1).

As with the human data, we also observed a higher proportion of read pairs mapping at longer intrachromosomal distances for DpnII CiFi versus the Hi-C dataset (Fig. 4A). Comparing read mapping similarly shows improved representation across unique space, intercalary heterochromatin, pericentromeric heterochromatin, putative centromeres, and repeat rich heterochromatin blocks for CiFi compared with Hi-C (Fig. 4B and Supplementary Tables 6 and 7). We next generated contact matrices for both CiFi (104 million contacts) and Hi-C (64 million contacts) datasets (Fig. 4C) and observed high correlation of overall contact signals at 2.5-Mbp resolution ($r^2 = 0.89$) (Supplementary Table 8). Comparisons of both contact matrices with another published high-coverage Hi-C dataset (125 million contacts)[26] of pooled *An. coluzzii* embryos showed consistent signals. For example, despite having reduced coverage in the CiFi dataset produced from a single male sample (Supplementary Table 7), we detected a chromosome X interaction-enriched signal between 7.5 Mbp and 15 Mbp, likely representing a previously validated X-loop configuration[26]. Together, these results show the ability of CiFi to scale down from millions to tens of thousands of cells, enabling improved mappings of reads across repetitive regions in samples with small starting materials (e.g., single small organisms or isolated cells) without sacrificing the ability to detect chromatin interactions.

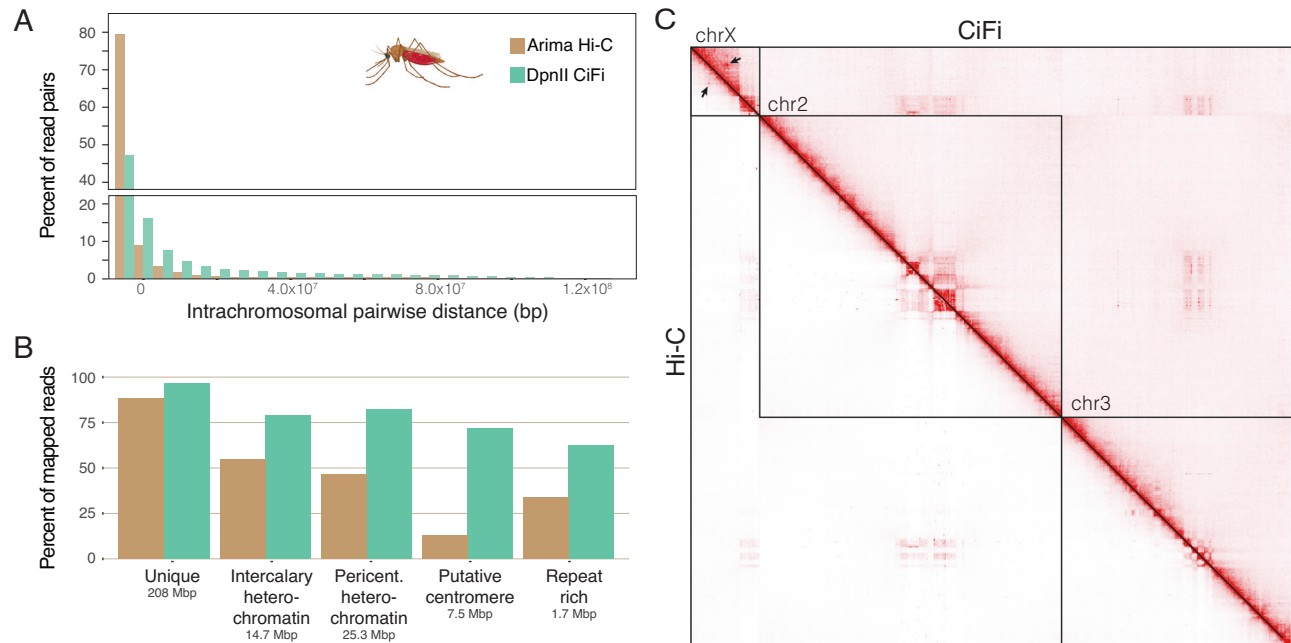

**Fig. 4 | 3C analysis of a single *Anopheles coluzzii* mosquito. A** Histogram of pairwise intrachromosomal distances between DpnII CiFi segments from the same HiFi read and Hi-C (Arima v2) paired reads plotted as a percent of total read pairs mapping intrachromosomally. **B** Percent of all mapped reads with MAPQ score of one or higher for Hi-C with Illumina and CiFi (DpnII) across different repetitive genome classifications (intercalary heterochromatin, pericentromeric (pericent.) heterochromatin, putative centromeres, and repeat rich regions; Supplementary

Table 6) with total lengths of each classification type across the genome indicated. Percent of mapped reads and normalized coverage at varied MAPQ score cutoffs can be found in Supplementary Fig. 12. **C** Genome-wide pairwise interaction maps (2.5-Mbp resolution) for CiFi (above diagonal) and Arima Hi-C (below diagonal), normalized using the Knight–Ruiz algorithm[61]. Arrows indicate the site of likely chromosome X looping. Mosquito icon created in BioRender. Dennis, M. (2025) https://BioRender.com/677ba91. Source data are provided as a Source Data file.

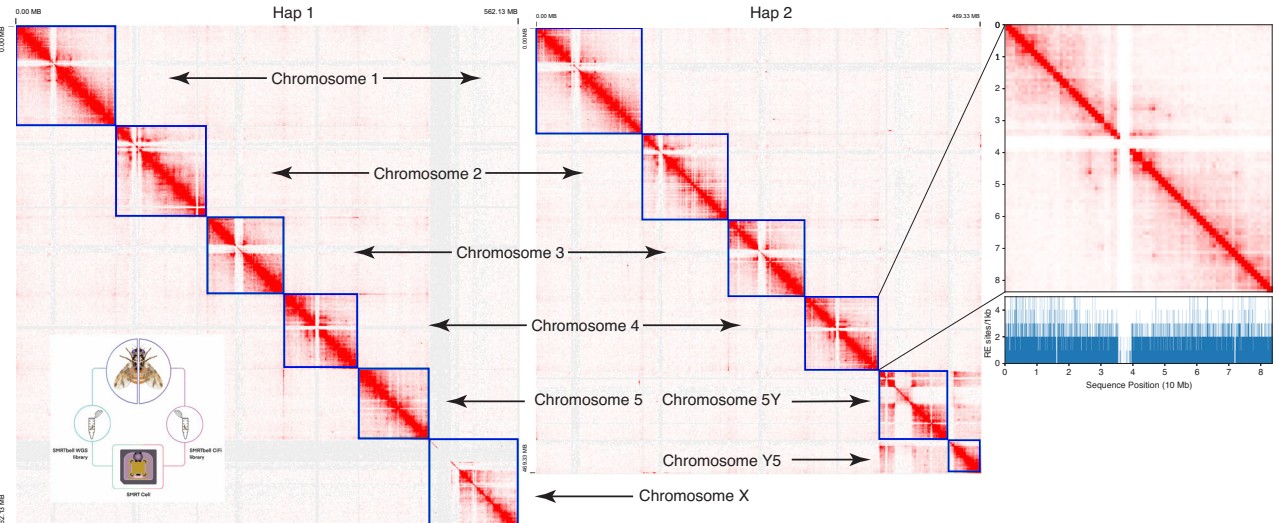

**Fig. 5 | De novo phased assembly of a single *Ceratitis capitata* Mediterranean fruit fly using a combination of HiFi and CiFi sequencing.** CiFi contacts were used to scaffold the HiFi contig assembly into a chromosome-scale, phased diploid assembly (shown separately for both haplotypes (Hap1 & Hap2)). Contacts were visualized using Juicebox using a MAPQ cutoff >0, 1-Mbp resolution, and Balanced ++ normalization. Inset: workflow for generating a HiFi whole-genome

sequencing and CiFi library from the same individual and subsequently combined for the same sequencing run. Outset: representative chromosome and occurrence of HindIII restriction sites (blue) across the chromosome. Lack of restriction enzyme (RE) sites explains the paucity of contacts in the central region. Source data are provided through NCBI BioProject accession PRJEB83708 and as a Source Data file.

## De novo diploid assembly of a single insect

We leveraged CiFi to assemble the ~600 Mbp genome of a single Mediterranean fruit fly, *Ceratitis capitata*, by generating PacBio HiFi whole-genome sequencing (WGS) and CiFi reads from opposite halves of the same male individual split laterally (Fig. 5 and Supplementary Table 9). In splitting the sample, half of the tissue yielded 1.35 μg of

high-molecular-weight DNA that was used for SMRTbell library preparation for HiFi WGS data generation. The post-crosslink reversal-purified DNA from the other half of the tissue was quantified at 1.11 μg and used for CiFi library generation. We successfully tested both the combining of HiFi and CiFi libraries in a single Revio sequencing run, and the barcoding and pooling of several CiFi libraries from different

samples ahead of sequencing (Supplementary Data 1). Using HindIII CiFi reads to both phase within[27] and scaffold across contigs[28] resulted in a chromosome-scale, diploid assembly ($2n = 12$) (Fig. 5). Adjusted QVs of the phased, chromosome-scale scaffolds ranged from 41.4 to 58.1[29] with 96.7–99.3% complete Diptera BUSCO values[30] (Supplementary Table 9).

Mapping CiFi fragments as paired-end reads back to the assembly showed the expected intrachromosomal contacts, and interactions between chromosomes 5 and Y, marking a known translocation characteristic of the strain used here[31]. Subsampling CiFi reads used to scaffold this assembly revealed that 70,000 CiFi reads, resulting in approximately 300,000 proximity-ligated pairs (equivalent to 1.5× coverage), produced a maximum theoretical L90 and chromosome-scale scaffolding quality that is on par with the curated reference[32] (Supplementary Fig. 13). Notably, subsampling to as low as 70,000 proximity-ligated pairs still achieved a near chromosome-scale scaffolded assembly compared to ~300,000 Hi-C interactions (Supplementary Fig. 14). Phasing and scaffolding using the NlaIII CiFi library yielded comparable results (Supplementary Table 9). These results demonstrate that not only can CiFi data be generated from low-input materials, in this case a single fly, but also the characteristics of the library are similar enough to standard HiFi libraries so that they can be pooled onto a single sequencing run, with the resulting data generated more than sufficient for synchronized characterization of the underlying genome and the three dimensional structure of the nuclei being processed. In this example, the CiFi data was superior to traditional Hi-C, with less CiFi data required to generate a chromosome-scale assembly (Supplementary Figs. 14 and 15). In addition, the CiFi data were more informative in heterochromatic regions, e.g., allowing the detailed characterization of a translocation associated with the Y chromosome, which has a complex and repetitive structure that short (traditional) Hi-C reads do not uniquely map to and thus fail to provide contact evidence in these regions (Supplementary Fig. 16).

### Limitations of the study and future directions

Highlighting limitations of the current CiFi approach, the use of restriction endonucleases has the potential to introduce biases in the regions assayed with certain loci naturally depleted for cutting sites; this can be mitigated by choosing alternative restriction sites or employing non-specific endonucleases[33,34], representing lines of future improvements. In addition, while we show that CiFi enables haplotype separation, phasing, and assembly scaffolding, improved computational approaches[35] that take full advantage of the higher-order multi-contact interactions inherent in CiFi reads can likely further enhance the resolving power of the approach. For example, there is room for improvement in the read mapping method that might leverage the positions of adjacent segments within the same CiFi read to improve confidence and accuracy in mappings even further.

In summary, CiFi enables efficient PacBio HiFi sequencing of 3C libraries from different species and sample types, allowing for high-quality, haplotype-resolved, chromosome-scale de novo genome assemblies with data from one sequencing technology, and, if desired, from a single sequencing run. A typical ~8 kbp CiFi read comprises multi-contact segments ranging in size from ~350 bp to 2 kbp that provide genome-wide chromatin contexts, including for highly repetitive and low-complexity regions. Its improved abilities to impute phasing will be powerful in future work characterizing haplotype-specific chromatin interactions and understanding allele-specific genome organization of imprinted regions. We anticipate that the low-input requirement will expand the application of 3C with long-read sequencing to many single small organisms, as well as other samples, including isolated cell types and disease specimens such as tumor biopsies.

## Methods

We developed an optimized protocol for multi-contact, long-read Hi-C library preparation with low DNA input, enabling efficient and accurate sequencing on the PacBio platform. The CiFi approach overcomes traditional high-input limitations and enhances the resolution of 3D chromatin interactions. For detailed, step-by-step procedures, including reagent volumes and amendments to protocols, please refer to the Supplementary Information and Supplementary Data 3, respectively. This research complies with all relevant ethical regulations and has been found to be human subjects research exempt by the University of California, Davis Institutional Review Board.

### Cell culture and cross-linking

The GM12878 cell line was donated by a Utah resident with ancestry of Northern and Western Europe (CEPH cohort) and obtained from the NIGMS Human Genetic Cell Repository at the Coriell Institute for Medical Research. GM12878 was cultured in RPMI 1640 medium supplemented with 15% fetal bovine serum and 1% penicillin-streptomycin. Cells were maintained at 37 °C in a $CO_2$ incubator and tested at least biannually for mycoplasma contamination using a PCR detection kit (abm, Cat# G238). For cross-linking, 5–10 million cells were washed three times with cold phosphate-buffered saline (PBS) and resuspended in PBS containing 1% formaldehyde (EMD Millipore, Cat# 818708). Following a 10-min incubation at room temperature, glycine was added to a final concentration of 125 mM to quench the reaction. Cells were incubated for an additional 5 min at room temperature and 10 min on ice, then centrifuged at $500 \times g$ for 5 min at 4 °C. The pellet was washed with PBS, snap-frozen in liquid nitrogen, and stored at −80 °C.

### Restriction enzyme digestion

The cell pellet was resuspended in 50 µL of protease inhibitor cocktail (Sigma Aldrich, Cat# P8340) and 500 µL of cold permeabilization buffer (10 mM Tris-HCl, pH 8.0; 10 mM NaCl; 0.2% IGEPAL CA-630). After a 15-min incubation on ice, cells were centrifuged and resuspended in 300 µL of chilled 1.5× digestion reaction buffer compatible with the chosen restriction enzyme (NEB). Chromatin was denatured by adding SDS to a final concentration of 0.1% and incubating at 65 °C for 10 min with gentle agitation. SDS was quenched with Triton X-100 to a final concentration of 1%. Permeabilized cells were digested with 1 U/µL of DpnII or HindIII (NEB) in a final volume of 450 µL, maintaining a 1× digestion buffer concentration. The mixture was incubated at 37 °C for 18 h with gentle mixing to prevent condensation inside the tube.

### Proximity ligation and reverse cross-linking

Following digestion, the restriction enzymes were heat-inactivated at 65 °C for 20 min for DpnII and chemically inactivated for HindIII with a final concentration of 0.1% (w/v) SDS at 65 °C for 20 min with 300 rpm rotation, followed by quenching with a final concentration of 1% v/v Triton X-100 (Sigma-Aldrich, Cat# 93443). The samples were then immediately placed on ice. T4 DNA ligase (NEB M0202L) and ligation buffer were added to the mixture, which was then incubated at 16 °C for 6 h with gentle rotation. To degrade proteins and reverse cross-links, Proteinase K (Thermo Fisher Scientific, Cat# 25530049), SDS, and Tween-20 were added. The sample was incubated at 56 °C for 18 h with intermittent mixing. DNA was purified using phenol-chloroform extraction and ethanol precipitation, and then resuspended in TE buffer. To evaluate experimental efficiency, undigested, digested, and ligated DNA products were analyzed via agarose gel electrophoresis (Supplementary Fig. 17).

### Size selection and quality control

For libraries prepared using DpnII, size selection was performed using AMPure PB beads (PacBio) at a 0.45× ratio according to the

manufacturer's protocol. DNA size distribution was assessed using Femto Pulse automated pulsed-field capillary electrophoresis, confirming an expected size range suitable for SMRTbell library preparation (Supplementary Fig. 18A). For HindIII, additional DNA shearing was required before size selection due to the longer fragment sizes. The DNA was sheared to approximately 10 kbp using a g-TUBE (Covaris, 520104), following the manufacturer's protocol, and then followed by size selection with AMPure PB beads (PacBio) at a 0.45× ratio. The DNA size distribution was similarly confirmed using Femto Pulse analysis.

### SMRTbell library preparation from low DNA input

Single-strand overhangs were removed by treating the DNA with DNA Prep Buffer, NAD, DNA Prep Additive, and DNA Prep Enzyme, followed by incubation at 37 °C for 15 min. DNA damage was repaired using DNA Damage Repair Mix v2 (PacBio) with a 30-min incubation at 37 °C. End repair and A-tailing were performed by adding End Prep Mix and incubating at 20 °C for 30 min and then at 65 °C for 30 min. Adapters were ligated to the repaired DNA using diluted Amplification Adapters (PacBio), Ligation Mix, Ligation Additive, and Ligation Enhancer, with incubation at 20 °C for 60 min. The SMRTbell library was purified using SMRTbell beads (PacBio) and eluted in EB buffer. Library concentration was measured using the Qubit dsDNA HS assay (Thermo Fisher Scientific).

### Library amplification via PCR

To amplify the library, PCR reactions were set up using KOD Xtreme hot-start polymerase (Sigma, Cat# 71975-M) due to its high fidelity and efficiency with low DNA input samples. Each reaction contained 2× Xtreme buffer, 2 mM dNTPs, sample amplification PCR primer, purified SMRTbell library, and polymerase. The PCR conditions were as follows: initial denaturation at 94 °C for 2 min; 13 cycles of denaturation at 98 °C for 10 s, annealing at 60 °C for 30 s, and extension at 68 °C for 10 min; and a final extension at 68 °C for 5 min. PCR cycle numbers were optimized based on initial template concentration to achieve the desired yield. Amplified DNA was purified using SMRTbell beads and quantified using the Qubit dsDNA HS kit. DNA size distribution was confirmed with Femto Pulse electrophoresis, ensuring a size mode of 8–10 kbp (Supplementary Fig. 18B).

### Final library preparation and sequencing

The amplified DNA underwent a second round of DNA damage repair, end repair, A-tailing, and adapter ligation using Overhang Adapter v3 or barcoded adapters for multiplexing (Supplementary Fig. 18C). The final SMRTbell library was purified using SMRTbell beads and assessed for concentration and size distribution. Size selection was performed using the BluePippin system (Sage Science) to enrich for fragments larger than 5 kbp for GM12878 DpnII (Supplementary Fig. 18D) and larger than 8 kbp for the Mediterranean fruit fly. For GM12878 with HindIII and mosquito, an alternative size selection method with diluted AMPure PB beads was performed. The initial optimization of the CiFi protocol for GM12878 with DpnII used the SMRTbell 2.0 prep kit. CiFi libraries for GM12878 with HindIII, mosquito, and Mediterranean fruit fly were used with the SMRTbell 3.0 prep kit. Size-selected libraries were sequenced either on PacBio Sequel II or Revio platforms for 24 h or 30 h (Supplementary Data 1).

### Modifications for cell titration CiFi library preparation

To accommodate samples with limited cell numbers, the standard Hi-C library preparation protocol was proportionally scaled down (Supplementary Data 3). For inputs of 5 million cells, reagents and volumes were reduced to 50% relative to the original approach for 10 million cells. For inputs between 1 million and 500,000 cells, volumes were reduced to 20%, and for inputs between 250,000 and 62,500 cells, a 10% scale-down was applied. These adjustments enabled efficient library preparation while maintaining protocol integrity.

### Modifications for CiFi library preparation using insect tissue

For preparation of a single adult male mosquito, *An. coluzzii* strain Ngousso, the above-described CiFi protocol was performed with minor modifications in size-selection steps. These included a 2× bead:sample SPRI select purification of following proximity ligation and reverse cross-linking, and 0.45:1 bead:sample ratio of AMPure PB beads prior to SMRTbell library preparation using the SMRTbell 3.0 prep kit.

To prepare a CiFi library from a single adult male Mediterranean fruit fly *C. capitata*, strain CDFA-Waimanalo (California Department of Food and Agriculture), the following modifications to the above protocol were made (Supplementary Data 3). Cross-linking (fixation) was performed using a 2% formaldehyde solution and centrifugations were performed at $1500 \times g$ for 5 min. Cross-link reversal was performed using 100 μL 20 mg/mL Proteinase K, 100 μL of 10% SDS, 72 μL of 5 M NaCl, and 728 μL NFW and DNA isolated using 1.8× SPRI-select purification. To minimize the presence of small fragments after proximity ligation and cross-link reversal but before gDNA amplification and library preparation, the isolated DNA was size selected using a 35% dilution of AMPure PB beads at 3.1×, to remove fragments smaller than 5 kbp. After SMRTbell library preparation using the SMRTbell 3.0 prep kit, the final library was size-selected using the BluePippin to remove fragments smaller than 8 kbp.

### CiFi sequence data processing

The CiFi sequence data were processed using the Pore-C Nextflow v23.04.3 workflow[8] with minor modifications made to account for differences in PacBio versus nanopore reads. Briefly, we performed an in silico digest using the Pore-c-py v2.0.6 Python package to split CiFi concatemer reads at known cut sites of the designated enzyme. The resulting segments were aligned to a reference genome (T2T-CHM13v2.0 for human LCL GM12878 and AcolN3 for *An. coluzzii*) using minimap2 v2.26[36] for PacBio reads (-x "map-hifi"). A mock Illumina-like paired-end BAM was generated from CiFi segments within each HiFi read, representing all possible paired interactions within the HiFi read. A filtering tool was developed to retain only paired-end reads where both mates are mapped, exceed a MAPQ threshold, and display proper pairing flags, eliminating orphaned reads. Published Pore-C reads representing GM12878 DpnII (GenBank SRA accessions: SRR11589389, SRR11589400, SRR11589403, SRR11589404, SRR11589405, SRR11589406) and GM12878 HindIII (GenBank SRA accessions: SRR11589407 and SRR11589408) were processed with the same Pore-C Nextflow v23.04.3 workflow with differences to account for nanopore reads.

### Hi-C sequence data processing

Published paired-end Illumina reads representing GM12878 DpnII Hi-C[13] (GenBank SRA accessions: SRR1658583, SRR1658588, and SRR1658596) were mapped to T2T-CHM13v2.0 and processed using Juicer (https://github.com/aidenlab/juicer/blob/main/CPU/juicer.sh)[37]. We note that the MboI enzyme was used in the originally published Hi-C experiments, which has the same sequence motif and methylation sensitivity as DpnII. Briefly, paired-end Illumina reads were mapped to a reference genome with bwa v0.7.17-r1188. The Juicer pipeline was subsequently used to remove PCR duplicates, assign positions of chimeric reads using restriction-site positions for DpnII, and generate a contact matrix for all reads. Previously generated *An. coluzzii* Hi-C sequencing data generated using Arima Hi-C kit v2 (GenBank SRA accessions: ERR9356762 and ERR9356763) were combined and processed using the standard Arima Genomics Hi-C bioinformatic pipeline (https://github.com/ArimaGenomics/mapping_pipeline/blob/master/

arima_mapping_pipeline.sh) following identical steps and software versions as recommended in the protocol.

### Haplotype phasing of CiFi and Hi-C datasets

Using WhatsHap v2.8 haplotag and phased variants derived from a near-T2T diploid assembly of GM12878, respectively to T2T-CHM13v2.0[14,15], CiFi and Hi-C read mappings (BAM file MAPQ > 0) were assigned haplotypes (HP:i:1 or HP:i:2) with the following options: "--ignore-read-groups", "--skip-missing-contigs", "--ignore-linked-read." Based on recommendations from previous work[5], we imputed haplotype to paired reads within 30 Mbp for any Hi-C interaction. Similarly, we applied this same principle to CiFi segments that have 100% of phased segments within 30 Mbp on the same CiFi read sharing the same haplotype.

### Chromatin pairwise interaction analyses

Contact matrices were created using de-duplicated BAM files of CiFi and Hi-C datasets together with pairtools v1.1.2[38] and JuicerTools v1.9.9[37]. All contact matrices depict read pairs with a MAPQ score of one or higher. Chromatin contact matrices were visualized using Juicebox v1.11.08[39] and the UCSC Genome Browser. Chromatin contact counts at 2.5-Mbp and 50-kbp resolutions were extracted from CiFi and Hi-C contact matrices using JuicerTools v1.9.9 dump with Knight–Ruiz normalization, and Pearson correlation coefficients were calculated using R package tidyverse. TAD domains were called using TopDom v0.10.1[17], and MoC[19] was used to compare between CiFi vs. Hi-C and Pore-C genome wide. Jaccard index was calculated using BEDTools v2.26.0 jaccard using two metrics: (1) any overlapping TADs, and (2) requiring at least 25% reciprocal overlap between two TADs. As a result of a 27-Mbp human satellite in the pericentromeric regions (HSat), Knight–Ruiz normalization fails for chromosome 9. The MoC and Jaccard index for chromosome 9 is based on non-normalized contact counts extracted with the JuicerTools v1.9.9 dump command.

### Mosquito repetitive regions

To define heterochromatin blocks, we evaluated the density of repeats annotated with EarlGrey v.5.1.1[40] in 10-kbp windows and checked against self-similarity dotplots generated with StainedGlass v.0.6[41] and visualized in HiGlass v.1.13[42]. Regions of increased repeat density agreed well with heterochromatin boundaries mapping data using previously published annotations of *Anopheles gambiae* (AgamP3)[43] and BLAST[44] to locate the markers in AcolN3.

To identify putative centromeres, we used tandem repeat annotations generated with ULTRA v.1.1.0[45], again combined with StainedGlass self-similarity dotplots visualized in HiGlass. Using the multiple centromere-associated repeats identified in *An. gambiae*[46], we defined the outermost coordinates of multi-component tandem-repeat blocks to define centromere locations. AcolN3 annotations of repetitive genomic regions can be found in Supplementary Table 6.

### Phased diploid genome assembly of a single insect

To demonstrate the application of CiFi to genome assembly of samples with limited tissue, a single Mediterranean fruit fly, *C. capitata*, was used to generate a uniquely barcoded PacBio HiFi low-input SMRTbell WGS library using the SMRTbell 3.0 prep kit and a uniquely barcoded CiFi library, which allows flexibility in sequencing across different SMRT Cells or on one. A single *C. capitata* male was divided laterally, where half of the tissue (~10 mg) was prepared into a CiFi library using HindIII and the insect CiFi library preparation protocol described above. Its corresponding other half was prepared into a SMRTbell WGS library using methods described for chromosome-scale tephritid genome assemblies[47,48]. Briefly, the insect tissue destined to be a HiFi WGS library was used to perform a

high molecular weight DNA extraction and sheared using the Diagenode Megaruptor 3, targeting a mean fragment size of 20 kb. A SMRTBell Express Template kit was used to prepare the HiFi library. The HiFi and CiFi libraries were sequenced on separate Revio SMRT Cells, where the CiFi library was pooled with CiFi libraries prepared for other species (representing about 12.5% of the run) and the WGS library was sequenced in a separate SMRT Cell by itself. A separate single *C. capitata* male was similarly divided, where half of the tissue was prepared into a CiFi library using NlaIII, and its corresponding other half was prepared into a WGS library. These two libraries were sequenced on one Revio SMRT Cell, pooling to approximate 80% of the data coming from the WGS library and 20% from the CiFi library.

For comparison, a Hi-C library was prepared using a separate individual using a previously published method[49]. Following that method, frozen tissue was fixed and digested with DdeI and DpnII. After digestion, proximity ligation was performed, crosslinks reversed, and DNA purified. The purified DNA was sheared with a Diagenode Bioruptor Pico and fragments isolated at sizes 200–600 bp. The size-selected DNA was used for short-read library generation using the NEB Next Ultra II DNA Library Prep Kit and followed by the Element Biosciences Adept Library Prep Kit. The final Hi-C library was sequenced on a portion of a 150-bp paired-end flow cell on an Element Biosciences Aviti instrument. This Hi-C data was used to assemble both Hap1 and Hap2 of the phased HiFi contig assembly by mapping Hi-C reads filtered for PCR duplicates to the Hap1 and Hap2 reference assemblies separately using the YaHS pipeline[28]. Hi-C contacts were visualized using Juicebox using the same resolution, normalization, and MAPQ threshold as the CiFi contacts.

The resulting HiFi and CiFi reads were used to create a separate single individual assembly for each CiFi/HiFi combination from each restriction enzyme. A custom script was used to create paired CiFi reads in silico (https://github.com/sheinasim-USDA/CiFi2PE), and these reads were supplied to Hifiasm v.0.24.0-r702 and the "--dual-scaf" option to phase the assembly of each autosome and both sex chromosomes using the graph structure of both phases[27]. Assessment of genome completeness was performed using BUSCO v.5.7.1[30] to identify the presence of the 3285 genes in the Diptera v10 database (diptera odb10). Raw and adjusted quality values representing base accuracy of the contig assemblies were calculated using YAK v0.1[29]. The contig assemblies were scaffolded with the matching individual CiFi reads using the Pore-C bioinformatic pipeline for desegmentation, mapping, and read pairing and YAHS v.1.2a.2 for scaffolding using the resulting read pairs[28]. Contig and scaffold statistics were calculated using the stats.sh function of BBMap v35.85[50]. Subsequently, one of the haplotypes of the HindIII assembly was scaffolded using a downsampling of the corresponding CiFi reads. Resulting L90s for assemblies scaffolded with 10,000 to 1,200,000 CiFi reads (~40,000 to ~4,600,000 proximity ligated pairs) were compared (Supplementary Fig. 13). Resulting contact maps for assemblies scaffolded with ~10,000 to ~1,000,000 CiFi proximity ligated pairs (Supplementary Fig. 14) and assemblies scaffolded with 10,000 to 1,000,000 Hi-C proximity ligated pairs (short-read pairs) were visualized using the same resolution, normalization, and MAPQ threshold described above (Supplementary Fig. 15).

### Statistics and reproducibility

PCR duplication rate was determined using pbmarkdup v1.1.0. Read-coverage comparisons were performed using Mosdepth v0.3.8[51] on the aligned 3C segments using 5-kbp windows with 1-kbp overlap between windows. Data was smoothed using a generalized additive model, and the resulting trend lines were compared between the standard 3C and CiFi sequence data. Distribution of fragment sizes and number of segments per read, including mean, median, mode,

and standard deviation for DpnII and HindIII, was quantified using R package tidyverse (dplyr v1.1.4, forcats v1.0.0, ggplot2 v3.5.2, lubridate v1.9.4, purrr v1.0.4, readr v2.1.5, stringr v1.5.1, tibble v3.2.1, and tidyr v1.3.1)[52]. The visualization of aligned segments of a CiFi read was performed using SVbyEye v0.99.0[53]. For the human reference T2T-CHM13v2.0, bedfiles were downloaded from UCSC Genome Browser representing different repetitive sequences, including SDs[54], interspersed repeats[55], cenSat[56], as well as unique sequences. For genomic hotspots, coordinates were converted from T2T-CHM13v1.0 to v1.1 from previously published[57,58] annotated regions using the UCSC liftOver tool and associated chain files obtained from https://s3-us-west-2.amazonaws.com/human-pangenomics/index.html?prefix=T2T/CHM13/assemblies/changes/v1.0_to_v1.1/. For the *An. coluzzii* reference AcolN3, repeat regions were defined for this study (see Methods; Supplementary Table 6). Bedfiles were intersected with aligned CiFi segments using BEDtools v2.26.0[59] and percentages of mapped segments were quantified at varying ≥ MAPQ 1, 10, 20, 30, and 60. These same bed coordinates were used to generate the coverage plots with mosdepth v0.3.8 for both human and *An. coluzzii* (Supplementary Figs. 5 and 12). No statistical method was used to predetermine sample sizes, and no data were excluded from the analyses. The experiments were not randomized. Investigators were not blinded to allocation during experiments and outcome assessment.

### Notes and considerations

Throughout the protocol, care was taken to prevent DNA shearing by using wide-bore pipette tips and avoiding vortexing. We recommend keeping the samples on ice at all times when not incubating, unless stated otherwise in the protocol. Quality control steps using Qubit assays and Femto Pulse electrophoresis were essential to monitor DNA concentration and fragment size at various stages. Reagent preparation and storage followed manufacturer recommendations to ensure enzyme activity and reaction efficiency. PCR cycle numbers were optimized based on DNA yield to produce sufficient library material for sequencing.

### Reporting summary

Further information on research design is available in the Nature Portfolio Reporting Summary linked to this article.

## Data availability

Raw and processed data generated from this study are available through NCBI BioProject and European Nucleotide Archive through accession PRJEB83708 or provided with this paper in the Supplementary Information/Source Data file. Previously published data used in this study are listed below. GM12878 Hi-C: SRR1658583 SRR1658588 SRR1658596 GM12878 Pore-C: SRR11589389 SRR11589400 SRR11589403 SRR11589404 SRR11589405 SRR11589406 SRR11589407 SRR11589408 *An. coluzzii* Hi-C: ERR9356762 ERR9356763 Platinum Pedigree Consortium [https://registry.opendata.aws/platinum-pedigree/] Source data are provided with this paper.

## Code availability

Bioinformatic pipelines are available https://github.com/mydennislab/CiFi and https://github.com/sheinasim-USDA/CiFi2PE and have been deposited at Zenodo (https://zenodo.org/records/17526543[60]).

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

## Acknowledgements

We would like to thank Dr. Noëlle Bittner for providing technical and administrative support and Karla Chávez-Congrains for assistance with illustrations. We also would like to thank Maryland Genomics at the University of Maryland School of Medicine—including Drs. Ellie Zhao, Lisa Sadzewicz, and Luke Tallon—for technical support in generating CiFi sequencing for human LCL samples. This work was supported, in part, by the U.S. National Science Foundation (CAREER 2145885 to M.Y.D.) and the National Institutes of Health (NIH) grants from the National Institute of Mental Health (R01MH132818 to M.Y.D.). This research used resources provided by the SCINet project and/or the AI Center of Excellence of the USDA Agricultural Research Service, ARS project numbers 0201-88888-003-000D and 0201-88888-002-000D and the Tropical Pest Genetics and Molecular Biology Research Unit ARS project number 2040-22430-028-000D. Mention of trade names or commercial products in this publication is solely for the purpose of providing scientific information and does not imply recommendation or endorsement by the U.S. Department of Agriculture. The USDA is an equal opportunity provider and employer. M.A.Q., A.M., and M.K.N.L. are supported by Wellcome through the 220540/Z/20/A award that supports the Wellcome Sanger Institute. Some images were created using BioRender.

## Author contributions

G.K., R.L.C., M.A.Q. performed the experimental studies. Computational analyses were carried out for the human LCL (S.P.M., M.A., and M.Y.D), mosquito (S.P.M., G.K, and A.M.), and the Mediterranean fruit fly (S.B.S.

and S.M.G). S.P.M., G.K., S.B.S., M.A.Q., S.M.G., M.K.N.L., J.K., and M.Y.D. wrote the initial manuscript with all authors contributing comments and edits. M.K.N.L., S.M.G., J.K, and M.Y.D. supervised the work.

## Competing interests

J.K. is a scientific advisor to and shareholder of Pacific Biosciences, a company developing single-molecule sequencing technologies. All other authors declare no competing interests.
