## [Transparent Peer Review file · Nature Communications]

CiFi: Accurate long-read chromosome conformation capture with low-input requirements

Corresponding Author: Dr Megan Dennis

Version 0:

Reviewer comments:

Reviewer #1

(Remarks to the Author)

McGinty et al present a new method, CiFi, to investigate 3-dimensional interactions in the genome using long-read sequencing via the PacBio platform. This results in an increase in the length of the sequenced fragments and in turn better mapping to low-complexity regions of the genome, such as transposable elements and centromeric regions. The method itself has the potential to have great impact in fields such as de-novo genome assembly, where all necessary sequencing data can be produced from one machine, and 3-dimensional organisation of eukaryotic genomes, where better resolution in areas previously missed by existing technologies could in the future be investigated more thoroughly. The study itself focuses on the former, generating a chromosome-scale assembly of a diploid fly from very little material and reads from PacBio machines. A lot of concepts and arguments demonstrating the efficacy and usefulness of CiFi are introduced, however these ideas are often not further developed to demonstrate the method fully, or present it in the context of other existing methods, such as Hi-C and PoreC. As the authors note, it is powerful to be able to produce these libraries from such small starting material, potentially in the tens of thousands of cells, however it will be difficult to know how much confidence people using this technique in the future can have without some more robust benchmarking in certain areas.

The elephant in the room from my reading of the manuscript was the lack of comparison with other long-read based 3C technologies, particular PoreC, which is hidden in a supplementary table. I understand there are inherent differences to the lab protocols, but given there are existing libraries using the same cell-line and enzymes (e.g. <https://www.ncbi.nlm.nih.gov/geo/query/acc.cgi?acc=GSE202539>), I think it would make sense to include a comparison correlating signal, coverage across low-complexity regions of the genome, and downsampling both technologies to the same number of reads/contacts/segments to help the reader.

As I understand it, the read segments are mapped to the genome independent of the other segments from the same read, so I assume differences in mapping to low-complexity regions stem from read length and potentially base quality of the reads. I wonder if the mapping could be made even more accurate by utilising the information that segments from the same read have a higher probability to map to the same chromosome and the mapping probability decreases with increased distance between each segment. An approach such as longranger's linked-read mapping for 10x reads might help in this regard.

What is the accuracy rate of these mappings to low-complexity regions? i.e. do all segments of the same CiFi read map to the same chromosome/scaffold/contig upon which the centromere/LINE element is found? I think it is not enough to quantify mapping accuracy purely based on the MAPQ score, I would like to see proportion of reads mapping to same chromosome from the same CiFi read for each segment of the genome (unique/centromere/etc) for the Hi-C and two CiFi libraries.

I find it a little difficult to reconcile the various coverage and interaction plots. In Figure 1B and S1 it appears that the signal is lost across the centromeric regions in the CiFi data, however the interaction matrix in 2C appears to contradict this. Furthermore I find Figure 2A difficult to interpret as it is not explained what the statistics are a percentage of. Is it reads mapping to these regions with a MAPQ greater than 1? I find this a little misleading without an indication of the total number of reads or coverage of each library mapping to these regions, which should also be included. A legend is also necessary to be able to interpret the fields in 2A.

The section including Topologically Associated Domains feels a little under-developed, and yet this has the potential to be one of the most interesting areas for a wider readership. This analysis should certainly be expanded to include the entirety of the human genome and comparisons in called TADs between CiFi, Hi-C and PoreC, which should be quantified using a metric such as a jaccard index. In particular I would be interested to see if new interacting domains or gene loops can be identified in areas of the genome which could not be investigated using other technologies, such as those low-complexity regions highlighted in the rest of the text.

It is claimed that chromatin interactions were successfully assayed using a DpnII CiFi library for *Anopheles coluzzii* with no indication of what this means. Results showing the correlation with a Hi-C library, coverage across each chromosome, repeat region, centromere should also be included in the results to help the reader interpret the outcomes.

Was the NlaIII library generated used for any analysis or de-novo assembly? This is only briefly mentioned in the methods and in the supplementary tables, but not in the results. Is the scaffolding better when using HindIII or NlaIII? Is the coverage different across the genome using the different enzymes? Does one obtain better results by combining the libraries to help phase/scaffold the genome assembly?

The test of downsampling of CiFi reads is interesting but difficult to contextualise without a comparison and an indication of accuracy. Are all of the scaffolds correct - i.e. map front to end to the curated reference genome? Is the scaffolding identical for all intermediate coverage levels and when does it fail? 1.5X already feels quite low, but does this still work at 0.5X or 0.01X? I would suggest including results detailing different coverage levels, number of structural variants between scaffolded assemblies and the reference and sizes of the scaffolds, either N50/N90 or an NX plot.

I also feel a similar test downsampling the number of Hi-C read-pairs or contacts, or even PoreC contacts, would be useful here to give the readers an indication of whether a similar number of contacts between the technologies results in different outcomes.

In Figure 3 there appear to be a number of regions in the de-novo assembly without any CiFi signal - what do these correspond to? Why for example is there a large gap in chr2 hap2 that is not in the hap1 chromosome? A more in-depth presentation of the coverage of the two haplotypes would help the reader to interpret the efficacy of using CiFi for haplotype phasing or phased genome assembly. Are the two haplotypes equally covered, is there evidence of switching errors between the two haplotypes?

It is not clear to me how the haplotype phased assemblies were created. Were the CiFi reads were added to the hifiasm command as hic reads? The default hap1 and hap2 assemblies from hifiasm cannot be considered "fully haplotype phased", as they do not utilise any long-range information, so can only phase on the contig level. The approach of generating phased scaffolds by using long-reads and Hi-C reads as input to the assembly is utilised by many groups as the default in generating diploid reference genomes. If there is a way to incorporate the CiFi reads into the assembly process, for example creating pseudo read-pair files from the in-silico digested file as with the read mapping, this would be very attractive to the genome assembly field. If this was already done, it should be made much clearer in the methods, ideally listing the commands, inputs and parameters, or including the full commands in the listed github entry.

Minor points:

A legend of the chromosome segments in Figure 1B would help the reader interpret the plot.

I think it would be helpful to also show the pairwise distance of the HindIII CiFi segments in Figure 1F

Overall, I think there is great merit to the method allowing for investigation of the 3-dimensional genome with higher resolution and producing all necessary data for a chromosome-scale de-novo genome assembly from a single sequencing run has the potential to be very powerful. I would however recommend more investigation and detail in the various results and claims throughout this manuscript.

Tom Brown
Leibniz-Institut für Zoo- und Wildtierforschung
Berlin Center for Genomics in Biodiversity Research
Königin-Luise-Straße 2-4 Gartenhaus
D-14195 Berlin

Reviewer #2

(Remarks to the Author)

This manuscript by McGinty and colleagues essentially introduces a improved protocol for 3C studies coupled to long-read sequencing. The authors improve the existing workflows on two aspects: on the output as regards increase in read length and in useful read percentage, and on the input material size by reducing cell numbers down to few tens of thousands. These are very welcome improvements, of course, but I see two major issues here. First, that all these improvements are quite incremental and heavily reliant on an already-published workflow for genomic DNA amplification (see doi: 10.1186/s13059-025-03487-9, wrongly cited as a preprint in the manuscript). This alone limits the technical advance of this work. Second, there is a strong focus on genome assemblies and phasing and very little exploration on what more this improved protocol might teach us about 3D genome folding. Therefore, although the description and presentation of the work is high quality, I am suggesting that, unless the latter issue on 3D genome folding is addressed by adding new insights

resulting from this improved protocol, the manuscript is likely better suited for a more specialized journal.

I typically disclose my identity to authors: A. Papantonis

Reviewer #3

(Remarks to the Author)

CiFi is a protocol that provides a real improvement in the efficiency of long-read multi-contact Hi-C and reduces the input requirements, making it possible to sequence single insects. I am supportive of publishing this paper in Nature Communications as long as the following issues are addressed.

Critiques:

Figure S5: There seems to be a small population of regions that has more contacts in DpnII Hi-C than in DpnII CiFi (Figure S5), above the diagonal. What is the nature of these regions? They are not made up of mostly centromere sequence as the ones enriched in CiFi.

Methods: The versions of packages used should be given in addition to the package names—this is inconsistent throughout the methods. The sequence data processing section should be more detailed, especially in terms of the aligner parameters.

The summary parameters are not enough to fully evaluate how well the libraries from low numbers of mammalian cells capture genome features. Please provide balanced contact matrices for the scaled-down GM12878 experiments.

Minor/Technical issues:

Page 3, Line 17: Isn't a pairwise interaction represented by a read pair? Therefore 136 pairwise interactions should correspond to 136 Illumina *read pairs* and 272 *individual Illumina paired end reads*.

Figure 2B: Why is the unscaled trans-chromosomal contact signal weaker in CiFi than in Hi-C?

Figure S6: The triangles are gray, not yellow as described in the caption.
How much DNA was obtained from each half of the fly? The amount of DNA is given for the mosquito but not the fly.

It is not entirely clear how the library columns in Table S3 correspond to the ones in Table S1.

Version 1:

Reviewer comments:

Reviewer #1

(Remarks to the Author)

The revised manuscript detailing the CiFi protocol and its applications in investigating genome topology and scaffolding and phasing of de-novo genome assemblies is in a much better shape. I really appreciate the extra details in the mapping quality and coverage of all technologies across the different regions of the genomes presented, as well as the more in-depth analysis of recovery and discover of TADS with the CiFi protocol compared to libraries from other technologies.

I am happy now that the claims made by the authors are well justified by the data presented.

I only have a couple of minor points:

Page 9 line 2: The quoted number of contacts should be 70,000, not 30,000. The value of 70k is stated in the rebuttal, but not the main text. One could also add "... compared to 300-500k Hi-C interactions", but I'll leave this to the authors.

Please add the <https://github.com/sheinasim-USDA/CiFi2PE> repository to the "Code Availability" section to improve findability.

(Remarks on code availability)

The tools and arguments are well detailed and will be useful to the community in analysing similar data or attempting to reproduce the steps in this paper.

Reviewer #2

(Remarks to the Author)

I find that the authors have done a commendable job in addressing critiques by all reviewers, and I fully support publication of their revised manuscript in Nat Commun.

(Remarks on code availability)

RESPONSE TO REVIEWERS

We thank all reviewers for their generally positive comments on the usefulness of the CiFi method, and their insightful suggestions to expand the scope of the manuscript to include additional comparisons/benchmarking and biological insights not previously possible using Hi-C. Accordingly, we have performed substantial additional analyses to address major concerns, including:

- Comparisons of CiFi with other 3C datasets including PoreC (*HindIII* and *DpnII*) for human lymphoblastoid cells (LCLs) GM12878 and Hi-C for the two insects.
- New results highlighting the significant improvements in haplotype phasing by leveraging multi-contact segments embedded in a single CiFi read using a published near-T2T genome of GM12878.
- Demonstrating significant improvements in TAD identification, namely across genomic hotspots regions enriched for segmental duplications, focusing specifically on the chromosome 1q21.1 disease-associated microdeletion/duplication region.
- Expanding the *Anopheles coluzzi* analysis to include new comparisons with published Hi-C datasets showing retained ability to discern biologically relevant chromatin structure using CiFi from a single mosquito while also demonstrating improvements in CiFi mapping qualities across repetitive regions.

We have provided individual comments below in point-by-point responses.

REVIEWER COMMENTS

Reviewer #1 (Remarks to the Author):

McGinty et al present a new method, CiFi, to investigate 3-dimensional interactions in the genome using long-read sequencing via the PacBio platform. This results in an increase in the length of the sequenced fragments and in turn better mapping to low-complexity regions of the genome, such as transposable elements and centromeric regions. The method itself has the potential to have great impact in fields such as de-novo genome assembly, where all necessary sequencing data can be produced from one machine, and 3-dimensional organisation of eukaryotic genomes, where better resolution in areas previously missed by existing technologies could in the future be investigated more thoroughly. The study itself focuses on the former, generating a chromosome-scale assembly of a diploid fly from very little material and reads from PacBio machines. A lot of concepts and arguments demonstrating the efficacy and usefulness of CiFi are introduced, however these ideas are often not further developed to demonstrate the method fully, or present it in the context of other existing methods, such as Hi-C and PoreC. As the authors note, it is powerful to be able to produce these libraries from such small starting material, potentially in the tens of thousands of cells, however it will be difficult to know how much confidence people using this technique in the future can have without some more robust benchmarking in certain areas.

We appreciate Reviewer 1's positive feedback and suggestions for further demonstrating that the CiFi method is robust based on benchmarking and comparisons to other methods. We note that the manuscript was originally formatted as a Correspondence Article with strict length limitations, which constrained the scope of our initial submission. In this revision, we have substantially expanded the study by adding two new figures and more comprehensive analyses, including direct comparisons between CiFi and Pore-C.

1. The elephant in the room from my reading of the manuscript was the lack of comparison with other long-read based 3C technologies, particularly PoreC, which is hidden in a supplementary table. I understand there are inherent differences to the lab protocols, but given there are existing libraries using the same cell-line and enzymes (e.g. <https://www.ncbi.nlm.nih.gov/geo/query/acc.cgi?acc=GSE202539>), I think it would make

sense to include a comparison correlating signal, coverage across low-complexity regions of the genome, and downsampling both technologies to the same number of reads/contacts/segments to help the reader.

We have added detailed comparisons between GM12878 *DpnII* and *HindIII* CiFi and the published Pore-C datasets of these same endonucleases, showing equivalent qualities (Figures 2C and S5) and coverage (Figure S5) of reads mapping across low-complexity regions of the human genome. We note that this analysis was performed without downsampling CiFi (e.g., 1.9 billion contacts for *DpnII*) to Pore-C (e.g., ~270 million contacts for *DpnII*), but still resulted in equivalent metrics with an $R^2 = 0.987$ when comparing CiFi and Pore-C contact signal (Table S4), genome-wide measure of TAD concordance MOC = 0.789 (Table S5), and a genome-wide TAD Jaccard index = 0.965 (Table S6). We have also included these findings in Results main text (p. 4, line 29):

Comparisons with published long-read Pore-C⁸ datasets for both *DpnII* and *HindIII* show equivalent mapping metrics with CiFi in percentage of reads mapped and coverage (Figures 2C, S4 and S5).

Figure 2. Comparisons of read mappings between short- and long-read 3C approaches. (C) Percent of all mapped reads with MAPQ score of one or higher for Hi-C with Illumina¹³ versus 3C with CiFi and Pore-C⁸ across different repetitive genome classifications (short interspersed nuclear elements (SINEs) *Alu* and *Mir*, long interspersed nuclear elements (LINEs) *L1* and *L2*, segmental duplications at both 90% (SD) and 98% (SD98) identity, and centromeres with and without the centromeric transition (CT) regions) with sizes of each type indicated. Percent of mapped reads and coverage at varied MAPQ score cutoffs can be found in Figures S4 and S5, respectively.

Figure S4. Mapping quality comparison of CiFi, Hi-C, and Pore-C reads across the genome. Percentage of reads (segments for CiFi and Pore-C) with varied MAPQ cutoffs (indicated above each plot) for Hi-C with Illumina (Rao et al., 2014), CiFi, and PoreC

(Deshpande et al. 2022) across different repetitive genome classifications (short interspersed nuclear elements (SINEs) *Alu* and *Mir*, long interspersed nuclear elements (LINEs) *L1* and *L2*, segmental duplications at both 90% (SD) and 98% (SD98) identity, and centromeres with and without the centromeric transition (CT) regions).

Figure S5. Normalized read coverage of CiFi, Pore-C, and Hi-C across the genome. Read coverage ratio (ratio of coverage per region to genomewide coverage) with varied MAPQ cutoffs (indicated above each plot) for Hi-C with Illumina (Rao et al., 2014), CiFi, and Pore-C (Deshpande et al. 2022) across different repetitive genome classifications (short interspersed nuclear elements (SINEs) *Alu* and *Mir*, long interspersed nuclear elements (LINEs) *L1* and *L2*, segmental duplications at both 90% (SD) and 98% (SD98) identity, and centromeres with and without the centromeric transition (CT) regions) divided by the genome-wide coverage for each library.

For *DpnII* CiFi and Pore-C libraries, we further generated and compared pairwise contact matrices through signal correlation and TAD concordance, finding higher correlations than with Hi-C for the latter. These results demonstrate that CiFi shows strong concordance with established long-read 3C technique Pore-C (albeit with that method's much larger DNA input requirements). We have added these analyses to the Results and Supplementary Figures and Tables (see below).

Results (p. 6, line 3):

Comparisons with *DpnII* Pore-C and CiFi shows even stronger correlation of contact matrices genome wide at 2.5-Mbp resolution ($r^2 = 0.93$; Tables S4–S6 and Figures S9 and S10) as well as concordance of identified TADs (MoC = 0.79). Together, CiFi, Pore-C, and Hi-C similarly represent genome organization across a majority of the human genome, with long-read approaches providing notable improvements across repetitive and complex regions.

Supplementary Information (Tables S4–S6 and Figures S9 and S10):

Figure S9. Comparison of chromatin contacts for human LCL GM12878 between DpnII CiFi and Pore-C genome wide and across chromosomes 1-5. Chromosome-scale pairwise interaction maps at 2.5-Mbp resolution for (A) genome wide (B) chromosome 1, (C) chromosome 2, (D) chromosome 3, (E) chromosome 4, and (F) chromosome 5. Red color scales with the number of paired reads per bin. Contact matrices are normalized using Knights-Ruiz algorithm. CiFi is above the diagonal while Pore-C is below.

Figure S10. Correlation of chromatin contact signal at 2.5 Mbp resolution between *DpnII* CiFi and Pore-C. $R^2 = 0.987$. These represent contact in 2.5-Mbp bins across the genome.

2. As I understand it, the read segments are mapped to the genome independent of the other segments from the same read, so I assume differences in mapping to low-complexity regions stem from read length and potentially base quality of the reads. I wonder if the mapping could be made even more accurate by utilising the information that segments from the same read have a higher probability to map to the same chromosome and the mapping probability decreases with increased distance between each segment. An approach such as longranger's linked-read mapping for 10x reads might help in this regard.

Thank you for this suggestion; while we agree that distance-based metrics will likely improve accuracy of mapping, we have opted to use existing methods published for a multi-contact 3C dataset (Pore-C) in order to appropriately compare our results with that method. We acknowledge that there is room for improvement in the bioinformatic approaches used to map and analyze these long-read chromatin contact maps and have included this as a limitation and future direction of our approach in the **Results: Limitations of the study and future directions** (p. 9, line 28):

In addition, while we show that CiFi enables haplotype separation, phasing, and assembly scaffolding, improved computational approaches³⁵ that take full advantage of the higher-order multi-contact interactions inherent in CiFi reads can likely further enhance the resolving power of the approach. For example, there is room for improvement in the read mapping method that might leverage the positions of adjacent segments within the same CiFi read to improve confidence and accuracy in mappings even further.

3. What is the accuracy rate of these mappings to low-complexity regions? i.e. do all segments of the same CiFi read map to the same chromosome/scaffold/contig upon which the centromere/LINE element is found? I think it is not enough to quantify mapping accuracy purely based on the MAPQ score, I would like to see proportion of reads mapping to same chromosome from the same CiFi read for each segment of the genome (unique/centromere/etc) for the Hi-C and two CiFi libraries.

A majority of segments, as shown previously for Hi-C, map to the same chromosome with expected differences between unique or repetitive regions, with CiFi generally higher. Interestingly, we observed limited differences between intrachromosomal mapping rates between unique (80.0%) and repetitive regions (79.3%) for CiFi data, in addition to between *DpnII* and *HindIII* CiFi results; the latter suggests that the gains in segment length

from using *HindIII* does not significantly impact the accuracy of the reads that are mapped. We have included this analysis in the Results (p.4, line 11) and as Figure S3.

Improvements in read mapping and phasing

We next mapped CiFi concatemer reads against the human reference genome (T2T-CHM13_v2¹²) and found that a majority of the segments map to homologous chromosomes for both *DpnII* and *HindIII*, a pattern consistent across both unique (80.0%) and repetitive (79.3%) regions Figures 2A and S3).

Figure S3. Proportion of segments within a CiFi read mapping to the same chromosome. The percentage of segments within a CiFi read mapping to the same chromosome, per segment, depicted as violin plots. The segments are characterized by the genomic bin that they share overlap with. Solid lines represent the mean for each region/restriction enzyme group with the gray line representing *DpnII* Hi-C average. Only the average was calculated for Illumina reads because analysis of paired ends produces a bimodal result (0% or 100%).

4. I find it a little difficult to reconcile the various coverage and interaction plots. In Figure 1B and S1 it appears that the signal is lost across the centromeric regions in the CiFi data, however the interaction matrix in 2C appears to contradict this. Furthermore I find Figure 2A difficult to interpret as it is not explained what the statistics are a percentage of. Is it reads mapping to these regions with a MAPQ greater than 1? I find this a little misleading without an indication of the total number of reads or coverage of each library mapping to these regions, which should also be included. A legend is also necessary to be able to interpret the fields in 2A.

Our goal in Figure 1B was to show that the CiFi approach, which includes a genome amplification step, does not result in any significant coverage dropouts when compared to the “Standard” non-amplified library prep approach (with no MAPQ cutoff). For this reason, we plotted read depth only across non-duplicated regions, excluding the centromere and other repetitive/duplicated regions that may suffer from spurious mapping. We

note that the Standard approach produced ~10x fewer reads from a single SMRT cell versus CiFi, which is why there is considerably more noise in that dataset. To make this clearer, we have amended **Figure 1B** to remove the centromere spanning data entirely and replaced it with an asterisk with clarifications of plotting only mappings in unique genomic space in the Figure 1 legend (underlined below).

The former Figure 2C (now **Figure 3B**) represents an interaction matrix of human chromosome 1, including all reads with MAPQ>0. While there is reduced read coverage across the centromere, these data were normalized using a Knight-Ruiz algorithm that takes into account variable read depth across the genome. We have updated the figure legend for improved clarity (underlined below):

The former Figure 2A (now **Figure 2C**) quantifies the percentage of reads with MAPQ>0 when considering all mapped reads (with no MAPQ filter) in each region type. We have similar plots in **Figure S4** at varying MAPQ cutoffs, showing duplicated regions have less high-quality reads mapping overall, but are higher for CiFi and Pore-C versus Hi-C. We have amended the legend to make this clearer, also including designations of the genomic region abbreviations.

and without the centromeric transition (CT) regions) with sizes of each type indicated. Percent of mapped reads and coverage at varied MAPQ score cutoffs can be found in Figures S4 and S5, respectively.

To address the reviewer's concern regarding coverage, we have included labels showing the amount of basepairs covered by each genomic bin in **Figure 2C** and have now also included a comparison of relative coverage across each repetitive region type (**Figure S5**).

5. The section including Topologically Associated Domains feels a little under-developed, and yet this has the potential to be one of the most interesting areas for a wider readership. This analysis should certainly be expanded to include the entirety of the human genome and comparisons in called TADs between CiFi, Hi-C and PoreC, which should be quantified using a metric such as a jaccard index.

We agree and have expanded our analysis of topologically associated domains (TADs) to the entirety of the human genome with comparisons between CiFi vs. Hi-C and Pore-C (**Tables S5 and S6**). Previously, this was quantified using Measure of Concordance (MoC). This metric assesses the overlap between pairs of TADs, measured in basepair number and overall TAD size. A value of 0 would indicate that TADs are completely discordant while 1 would represent perfect concordance. We have made this more clear in the **Results**. Also, per this suggestion, we have added the Jaccard index overall and one requiring TADs to have at least 25% reciprocal overlap to be compared. Overall, we find similar metrics between CiFi vs. Hi-C and Pore-C.

Results (p. 5, line 32)

We next used chromatin interactions to partition the genome into TADs¹⁶, or regions that interact with each other in 3D chromatin more often than adjacent regions¹⁷. Applying the measure of concordance (MoC) metric¹⁸—which assesses basepair overlap between domains considering overall chromosome size—finds high concordance between CiFi and Hi-C TADs genome wide (MoC = 0.83) (Tables S5 and S6 and Figure S8). Comparisons with *DpnII* Pore-C and CiFi shows even stronger correlation of contact matrices genome wide at 2.5-Mbp resolution ($r^2 = 0.93$; Tables S4–S6 and Figures S9 and S10) as well as concordance of identified TADs (MoC = 0.79).

In particular I would be interested to see if new interacting domains or gene loops can be identified in areas of the genome which could not be investigated using other technologies, such as those low-complexity regions highlighted in the rest of the text.

To address this comment (similarly raised by Reviewer 2), we have added new **Results** comparing “gaps” due to insufficient pairwise interactions identified in the TopDom TAD analysis, partitioning results genomewide and across SDs and centromeric regions. These results demonstrate that Hi-C consistently has more than twice the basepairs covered by TAD gaps 5% across the genome, 15% across SDs, and 34% across centromeric satellites, compared to 2%, 8.6%, and 18% from CiFi data, respectively. We performed the same analysis across disease- and phenotype-relevant genomic hotspots enriched for SDs finding similar enrichments of gaps in Hi-C vs CiFi. This prompted us to perform a deeper analysis of TADs across the chromosome 1q21.1 disease-associated locus. From this, we find that not only does the Hi-C TAD analysis produce mostly gaps across the region not seen using CiFi data, but CiFi also reveals the presence of a TAD “boundary” overlapping a disease relevant locus unseen in the Hi-C data (**Figure 3C**).

Results (p. 6, line 22):

To better understand the extent to which CiFi improves assessment of chromatin interactions, we cataloged genomic regions classified as “gaps” in the TAD analysis—representing loci with insufficient data to call domains. Gaps are found less frequently in TADs delineated using CiFi (~2% of human autosome basepairs) compared with Illumina Hi-C (~5%), with more marked differences across repetitive regions (SDs: 8.6% CiFi vs 18% Hi-C; centromeres: 15% CiFi

vs 34% Hi-C). These gaps present challenges in analysis of human genomic hotspots—regions enriched for SDs that carry some of the greatest amounts of structural polymorphism in the genome²⁰—where we observed none across flanking SDs using CiFi, in contrast to 8% using Hi-C.

One such locus is present on chromosome 1q21.1 where recurrent copy-number variants are associated with human developmental conditions^{21,22}, including intellectual disability, autism, congenital abnormalities, and thrombocytopenia absent radius (TAR) syndrome (Figure 3C). While Hi-C TAD analysis of this locus contains mostly gaps due to insufficient interaction data, CiFi TAD analysis shows no gaps. Even in regions where Hi-C TAD domains are identified, discrepancies exist, e.g. across the ~200 kbp TAR syndrome locus, Hi-C classifies the unique portion of the locus as a domain flanked by two gaps spanning the adjacent SDs mediating the disease-associated microdeletion. In contrast, CiFi assigns the locus as a TAD “boundary”, a region that serves as a barrier between adjacent TADs. While further work is necessary to validate this finding, boundaries have been shown to be enriched for active chromatin and highly expressed genes^{23,24}. Several housekeeping genes reside at this locus (*POLR3GL*, *POLR3C*, *RBM8A*), thereby supporting the CiFi result. The use of CiFi to characterize chromatin interactions at genomic hotspots such as this opens exciting new directions for understanding how structural variants impact gene regulation and underlying disease etiologies.

Figure 3. Pairwise chromatin contacts for human LCL GM12878. *DpnII* libraries for CiFi (above diagonal) and Hi-C (below diagonal) were used to generate pairwise interaction maps (A) genome wide (not normalized and at 2.5-Mbp resolution); and (B) across chromosome 1 (normalized using the Knight-Ruiz algorithm¹⁹ and at 250-kbp resolution). Cytogenetic bands and the centromere (criss-cross pattern) are depicted on each chromosome 1. Increasing red shading corresponds to greater paired interactions. Equivalent plots of CiFi and Pore-C can be found in Figure S9. (C) UCSC Genome Browser view of 1q21.1-q22 (chr1:142.3 Mbp-150.4 Mbp; T2T-CHM13v2) with TopDom¹⁶ topologically associating domain (TAD) classifications (“domain” in black, “boundary” in green, “gap” in tan), centromere satellites (cenSat), and segmental duplications (SDs) track depicted. Two genomic “hotspots” are delineated, including the disease-associated TAR syndrome locus (purple) and chromosome 1q21.1 deletion

(del) and duplication (dup) region (blue). TADs are also depicted as yellow triangles along the contact matrix. Pairwise interactions for CiFi are depicted on top and Hi-C on the bottom at 10-kbp resolution.

6. It is claimed that chromatin interactions were successfully assayed using a *DpnII* CiFi library for *Anopheles coluzzii* with no indication of what this means. Results showing the correlation with a Hi-C library, coverage across each chromosome, repeat region, centromere should also be included in the results to help the reader interpret the outcomes.

We have significantly expanded our analysis of the *Anopheles coluzzii DpnII* CiFi dataset by performing comparisons with a separate published dataset from another *An. coluzzii* individual generated with Arima Hi-C. The results are now represented as a new **Figure 4** and includes: (1) pairwise contact signal correlation between the two datasets; (2) comparison of intrachromosomal paired-read distances (**Figure 4A**); (3) assessment of read-mapping qualities (**Figure 4B**) and coverages (**Figure S12**) across different genomic regions (euchromatic regions and various heterochromatin regions: intercalary heterochromatin, pericentromeric heterochromatin, and centromeres). Further, we compared our results with a separate published study and identified an X-chromosome loop in both the CiFi and Illumina (**Figure 4C**), demonstrating that generating CiFi libraries with very low input material results in trustworthy results.

Results (p. 7, line 20):

Figure 4. Chromatin conformation capture analysis of a single *Anopheles coluzzii* mosquito. (A) Histogram of pairwise intrachromosomal distances between *DpnII* CiFi segments from the same HiFi read and Hi-C (Arima v2) paired reads plotted as a percent of total read pairs mapping intrachromosomally. (B) Percent of all mapped reads with MAPQ score of one or higher for Hi-C with Illumina and CiFi (*DpnII*) across different repetitive genome classifications (intercalary heterochromatin, pericentromeric (pericent.) heterochromatin, putative centromeres, and repeat rich regions; Table S8) with total lengths of each classification type across the genome indicated. Percent of mapped reads and normalized coverage at varied MAPQ score cutoffs can be found in Figure S12. (C) Genome-wide pairwise interaction maps (2.5 Mbp resolution) for CiFi (above diagonal) and Hi-C (below diagonal), normalized using the Knight-Ruiz algorithm¹⁹. Arrows indicate the site of likely chromosome X looping. CiFi is above the diagonal while Arima Hi-C is below.

Scaling down CiFi for smaller sample inputs

To explore the benefits of lower input requirements, we successfully scaled down GM12878 *DpnII* CiFi samples by over 100-fold, from 10 million cells (~60 µg of DNA input) to 62,000 cells (~370 ng), resulting in consistent

sequencing metrics and contact matrices across all starting amounts (Tables S1 and S7 and Figure S11). Moving beyond human cell lines, we next applied *DpnII* CiFi to a single male *Anopheles coluzzii* mosquito (~250 ng starting input²⁵) and generated 2.37 million HiFi reads and 21.1 million segments of median length 509 bp (Tables S1 and S3). We mapped these data across the 263-Mbp reference genome (AcolN3) and compared with Hi-C reads from a female *An. coluzzii* generated with two four-cutter restriction enzymes used to aid in the assembly of AcolN3 (NCBI RefSeq GCF_943734685.1).

As with the human data, we also observed a higher proportion of read pairs mapping at longer intrachromosomal distances for *DpnII* CiFi versus the Hi-C dataset (Figure 4A). Comparing read mapping similarly shows improved representation across unique space, intercalary heterochromatin, pericentromeric heterochromatin, putative centromeres, and repeat rich heterochromatin blocks for CiFi compared with Hi-C (Figure 4B and Tables S8 and S9). We next generated contact matrices for both CiFi (104 million contacts) and Hi-C (64 million contacts) datasets (Figure 4C) and observed high correlation of overall contact signals at 2.5-Mbp resolution ($r^2 = 0.89$) (Table S10). Comparisons of both contact matrices with another published high-coverage Hi-C dataset (125 million contacts)²⁶ of pooled *An. coluzzi* embryos showed consistent signals. For example, despite having reduced coverage in the CiFi dataset produced from a single male sample (Table S9), we detected a chromosome X interaction-enriched signal between 7.5 Mbp and 15 Mbp, likely representing a previously validated X-loop configuration²⁶. Together, these results show the ability of CiFi to scale down from millions to tens of thousands of cells, enabling improved mappings of reads across repetitive regions in samples with small starting materials (e.g., single small organisms or isolated cells) without sacrificing the ability to detect chromatin interactions.

Figure S12. Percentage of reads mapped and normalized read coverage of CiFi and Hi-C across the genome for *Anopheles coluzzii*. Shown on the left is percent of mapped reads at varied MAPQ cutoffs (indicated to the left of plots) of total mapped reads

(with no MAPQ cutoff) per region. for Hi-C with Illumina (Arima v2) and CiFi *DpnII* across different repetitive genome classifications (intercalary heterochromatin, pericentromeric (pericent.) heterochromatin, putative centromeres, and repeat rich regions; see Table S8) with total lengths of each classification type across the genome indicated. Shown on the right is read-coverage ratio (ratio of coverage per region to genomewide coverage) with varied MAPQ cutoffs for the same regions and sequencing libraries.

7. Was the *NlaIII* library generated used for any analysis or de-novo assembly? This is only briefly mentioned in the methods and in the supplementary tables, but not in the results. Is the scaffolding better when using *HindIII* or *NlaIII*? Is the coverage different across the genome using the different enzymes? Does one obtain better results by combining the libraries to help phase/scaffold the genome assembly?

There were no significant differences in scaffolding between the *HindIII* and *NlaIII* libraries. This is now mentioned in the **Results** (p. 9, line 4) and metrics can be found in **Table S11**.

8. The test of downsampling of CiFi reads is interesting but difficult to contextualise without a comparison and an indication of accuracy. Are all of the scaffolds correct - i.e. map front to end to the curated reference genome? Is the scaffolding identical for all intermediate coverage levels and when does it fail? 1.5X already feels quite low, but does this still work at 0.5X or 0.01X? I would suggest including results detailing different coverage levels, number of structural variants between scaffolded assemblies and the reference and sizes of the scaffolds, either N50/N90 or an NX plot.

I also feel a similar test downsampling the number of Hi-C read-pairs or contacts, or even PoreC contacts, would be useful here to give the readers an indication of whether a similar number of contacts between the technologies results in different outcomes.

As suggested by Reviewer 1, we generated Hi-C data from a male from the same strain as the HiFi/CiFi data and downsampled the Hi-C data to the same amount of proximity-ligated pairs to scaffold the contig assembly of hap1. This was visualized (**Figure S16**), along with analogous CiFi data (**Figure S15**), for increasing improvements to the scaffolding with increasing number of proximity-ligated pairs. Downsampling the CiFi data from the 300k pairs (or 1.5x coverage shown in **Figure S13** to achieve a maximum theoretical L90 of the final curated assembly) found that chromosome-scale scaffolding was achievable with ~70k proximity-ligated pairs (**Figure S14D and F**). Alternatively, Hi-C data needed a minimum of ~300k proximity ligated pairs (**Figure S15F**) and failed to achieve a comparable LD90 at 1M proximity ligated pairs (**Figure S15I**). Additionally, a comparison of the CiFi and Hi-C data for low-complexity regions that provide evidence of the translocation was visualized and included as **Figure S16**. We have also included these findings in our **Results, Supplementary Information, and Methods** (p.14, line 16).

Results (p.8, line 30):

Subsampling CiFi reads used to scaffold this assembly revealed that 70,000 CiFi reads, resulting in approximately 300,000 proximity-ligated pairs (equivalent to 1.5× coverage), produced a maximum theoretical L90 and chromosome-scale scaffolding quality that is on par with the curated reference³² (Figure S13). Notably, subsampling to as low as 30,000 proximity-ligated pairs still achieved a near chromosome-scale scaffolded assembly (Figure S14). Phasing and scaffolding using the *NlaIII* CiFi library yielded comparable results (Table S11). These results demonstrate that not only can CiFi data be generated from low-input materials, in this case a single fly, but also the characteristics of the library are similar enough to standard HiFi libraries so that they can be pooled onto a single sequencing run, with the resulting data generated more than sufficient for synchronized characterization of the underlying genome and the three dimensional structure of the nuclei being processed. In this example, the CiFi data was superior to traditional Hi-C, with less CiFi data required to generate a chromosome-scale assembly (Figures S14 and S15). In addition, the CiFi data was more informative in heterochromatic regions, e.g. allowing the detailed characterization of a translocation

associated with the Y chromosome, which has a complex and repetitive structure that short (traditional) Hi-C reads do not uniquely map to and thus fail to provide contact evidence in these regions (Figure S16).

Figure S14. Scaffolding the Mediterranean fruit fly assembly with varying coverages of CiFi segments. Subsampling paired segments of *HindIII* CiFi at (A) 10k read pairs, (B) 30k read pairs, (C) 50k read pairs, (D) 70k read pairs, (E) 100k read pairs, (F) 300k read pairs, (G) 500k read pairs, (H) 700k read pairs, and (I) 1000k read pairs produced varying number of chromosome scaffolds from YaHS.

Figure S15. Scaffolding the Mediterranean fruit fly assembly with varying coverages of Hi-C reads. Subsampling paired short reads (Element Biosciences Aviti) of Hi-C with *DdeI* and *DpnII* at (A) 10k read pairs, (B) 30k read pairs, (C) 50k read pairs, (D) 70k read pairs, (E) 100k read pairs, (F) 300k read pairs, (G) 500k read pairs, (H) 700k read pairs, and (I) 1000k read pairs produced varying number of chromosome scaffolds from YaHS.

9. In Figure 3 there appear to be a number of regions in the de-novo assembly without any CiFi signal - what do these correspond to? Why for example is there a large gap in chr2 hap2 that is not in the hap1 chromosome? A more in-depth presentation of the coverage of the two haplotypes would help the reader to interpret the efficacy of using CiFi for haplotype phasing or phased genome assembly. Are the two haplotypes equally covered, is there evidence of switching errors between the two haplotypes?

We agree that ideally our assembly would be benchmarked, but we do not have a truth data set, nor data from the parents, and thus are unable to validate the accuracy of the phasing. We believe the discrepancy in mapping between hap1 and hap2 in the previous visualization of the contacts is an artifact of the fact that the CiFi data was mapped to the diploid assembly and the mapping software favored one haplotype over the other, but visualization of the two haplotypes separately reveals no difference in mapping (**Figure 5**).

resolution, and Balanced ++ normalization. Inset: workflow for generating a HiFi whole-genome sequencing and CiFi library from the same individual and subsequently combined for the same sequencing run. Outset: representative chromosome and occurrence of *HindIII* restriction sites (blue) across the chromosome. Lack of restriction sites explains the paucity of contacts in the central region.

10. It is not clear to me how the haplotype phased assemblies were created. Were the CiFi reads were added to the hifiasm command as hic reads? The default hap1 and hap2 assemblies from hifiasm cannot be considered "fully haplotype phased", as they do not utilise any long-range information, so can only phase on the contig level. The approach of generating phased scaffolds by using long-reads and Hi-C reads as input to the assembly is utilised by many groups as the default in generating diploid reference genomes. If there is a way to incorporate the CiFi reads into the assembly process, for example creating pseudo read-pair files from the in-silico digested file as with the read mapping, this would be very attractive to the genome assembly field. If this was already done, it should be made much clearer in the methods, ideally listing the commands, inputs and parameters, or including the full commands in the listed github entry.

Our original diploid assembly was performed using this exact procedure (*in silico* digest of CiFi reads to make paired reads that were used as input into hifiasm). To make this point more clear, we have modified the **Methods** accordingly. Further, the script we used to perform this analysis is now publicly available on GitHub and is referenced in the manuscript.

Methods (p. 14, line 28):

The resulting HiFi and CiFi reads were used to create a separate single individual assembly for each CiFi/HiFi combination from each restriction enzyme. A custom script was used to create paired CiFi reads *in silico* (<https://github.com/sheinasim-USDA/CiFi2PE>) and these reads were supplied to Hifiasm v.0.24.0-r702 and the `--dual-scaf` option to phase the assembly of each autosome and both sex chromosomes using the graph structure of both phases²⁷. Assessment of genome completeness was performed using BUSCO v.5.7.1³⁰ to identify the presence of the 3285 genes in the Diptera v10 database (diptera odb10). Raw and adjusted quality values representing base accuracy of the contig assemblies were calculated using YAK²⁹. The contig assemblies were scaffolded with the matching individual CiFi reads using the Pore-C bioinformatic pipeline for desegmentation, mapping, and read pairing and YAHS v.1.2a.2 for scaffolding using the resulting read pairs²⁸. Contig and scaffold statistics were calculated using the stats.sh function of BMAP⁴³. Subsequently, one of the haplotypes of the *HindIII* assembly was scaffolded using a downsampling of the corresponding CiFi reads. Resulting L90s for assemblies scaffolded with 10,000 to 1,200,000 CiFi reads (~40,000 to ~4,600,000 proximity ligated pairs) were compared (Figure S13). Resulting contact maps for assemblies scaffolded with ~10,000 to ~1,000,000 CiFi proximity ligated pairs (Figure S14) and assemblies scaffolded with 10,000 to 1,000,000 Hi-C proximity ligated pairs (short read pairs) were visualized using the same resolution, normalization, and MAPQ threshold described above (Figure S15).

Minor points:

A legend of the chromosome segments in Figure 1B would help the reader interpret the plot.

Thank you for this suggestion. The colors on the chromosome 1 ideogram represent cytogenetic bands and the centromeric regions (striped & criss-cross pattern). We note our analysis focuses on "unique" regions of the genome, and have added an "*" to indicate that the centromere is a repetitive region (see response on p. 7 to comment #4)

I think it would be helpful to also show the pairwise distance of the *HindIII* CiFi segments in Figure 1F

We agree and have now added this to Figure 1F (now **Figure 2B**) as the Reviewer has suggested, with results largely matching those found for *DpnII* CiFi. For comparison purposes, we have also added *DpnII* Hi-C

pairwise distance metrics. At least, for these representative datasets, Both CiFi datasets result in a larger proportion of long-distance contacts versus Hi-C. This result was also recapitulated in the *An. coluzzi* analysis.

Results (p. 4, line 17):

Focusing on intrachromosomal pairwise interactions shows the expected 3C decay with increasing distance and spanning all length scales, going as far as >100 Mbp (the average size of a human chromosome; Figure 2B). Further, both *DpnII* and *HindIII* CiFi show proportionally more long-distance intrachromosomal interactions compared with a 101-bp paired-end Illumina Hi-C dataset generated as a standard resource for GM12878¹³.

Figure 2. Comparisons of read mappings between short- and long-read 3C approaches. (B) Histogram of pairwise intrachromosomal distances between *DpnII* and *HindIII* CiFi segments from the same HiFi read and *DpnII* Hi-C paired reads plotted as a percent of total read pairs mapping intrachromosomally.

Overall, I think there is great merit to the method allowing for investigation of the 3-dimensional genome with higher resolution and producing all necessary data for a chromosome-scale de-novo genome assembly from a single sequencing run has the potential to be very powerful. I would however recommend more investigation and detail in the various results and claims throughout this manuscript.

We appreciate the Reviewer's positive feedback and have aimed to provide a more thorough investigation of results.

Tom Brown
Leibniz-Institut für Zoo- und Wildtierforschung
Berlin Center for Genomics in Biodiversity Research
Königin-Luise-Straße 2-4 Gartenhaus
D-14195 Berlin

Reviewer #2 (Remarks to the Author):

1. This manuscript by McGinty and colleagues essentially introduces a improved protocol for 3C studies coupled to long-read sequencing. The authors improve the existing workflows on two aspects: on the output as regards increase in read length and in useful read percentage, and on the input material size by reducing cell numbers down to few tens of thousands. These are very welcome improvements, of course, but I see two major issues here. First, that all these improvements are quite incremental and heavily reliant on an already-published workflow for genomic DNA amplification (see doi: 10.1186/s13059-025-03487-9, wrongly cited as a preprint in the manuscript). This alone limits the technical advance of this work.

Thank you for pointing out the discrepancy in the reference. This was previously missed because the Bein et al. study had yet to be published in *Genome Biology* at time of our original submission. We have now updated the reference in the text. We additionally make reference to the published method, drawing attention to how our study demonstrates that the modified whole-genome amplification approach performs similarly well in 3C post-crosslink-reversal purified DNA as it does in the challenging genomic DNA samples presented by Bein and colleagues.

Results (p. 3, line 22):

Hypothesizing that residual cross-links remained on the DNA that prevent productive sequencing, following 3C we implemented a genome-wide amplification-based protocol designed for challenging samples¹¹, using a high-fidelity PCR enzyme to enrich for uncross-linked molecules ahead of sequencing (Figure 1A). This dramatically increased raw sequence yields and read lengths to standard sequencing performance (mean 105 kbp) and conversion to HiFi data (49.3% of polymerase reads; 30.6 Gbp at mean read length 9.35 kbp and median read quality value (QV) 38; Table S1). The published study¹¹ amplified genomic DNA from several organisms and observed limited PCR biases; this was also the case for the GM12878 *DpnII* 3C library with 1.8% of data representing PCR duplicates and no obvious dropouts evident when comparing read coverage with and without amplification (Figures 1B and S1).

Further, we respectfully disagree that the improvements between MC-3C and CiFi are incremental. Our method achieves >500-fold higher efficiency, enabling widespread adoption that was previously impractical. CiFi delivers significantly improved sequencing performance (5-10-fold higher) while requiring substantially less starting material (>100-fold reduction) compared to both long-read 3C approaches (MC-3C and Pore-C).

The limited adoption of MC-3C since its 2020 publication—with few/no subsequent published applications by other groups, to our knowledge—underscores the practical barriers our method addresses. In contrast, we already have active collaborations successfully applying CiFi across diverse organisms (insects, plants, and mammals), demonstrating its broader utility and accessibility. Publishing this improved protocol will make these capabilities available to the wider research community.

2. Second, there is a strong focus on genome assemblies and phasing and very little exploration on what more this improved protocol might teach us about 3D genoas-me folding. Therefore, although the description and presentation of the work is high quality, I am suggesting that, unless the latter issue on 3D genome folding is addressed by adding new insights resulting from this improved protocol, the manuscript is likely better suited for a more specialized journal.

To address Reviewer's 2 concerns, we have significantly expanded the scope of the study to include additional analysis comparing CiFi vs Hi-C datasets to: (1) discover topologically associating domains (TADs) across repetitive elements, highlighting disease-associated loci; and (2) perform haplotype phasing genome wide.

As described above in Reviewer 1 Response #5 (p.8), we have added focused analysis on improvements in assignment of TADs across repetitive regions (SDs and centromeric satellites) using CiFi vs Hi-C. Specifically, we find that more TADs can be delineated in disease- and phenotype-relevant genomic hotspots that are enriched for SDs. We highlight one such region on chromosome 1q21.1 with associations to several developmental defects and neurological conditions in a new **Figure 3C**. Using CiFi to characterize chromatin interactions, we specifically gain insights into the TAD structure for the 200-kbp thrombocytopenia absent radius (TAR) syndrome locus, finding that the region intersects a TAD “boundary” not reported by Hi-C data.

Results (p. 6, line 8):

Figure 3. Pairwise chromatin contacts for human LCL GM12878. *DpnII* libraries for CiFi (above diagonal) and Hi-C (below diagonal) were used to generate pairwise interaction maps (A) genome wide (not normalized and at 2.5-Mbp resolution); and (B) across chromosome 1 (normalized using the Knight-Ruiz algorithm¹⁹ and at 250-kbp resolution). Cytogenetic bands and the centromere (criss-cross pattern) are depicted on each chromosome 1. Increasing red shading corresponds to greater paired interactions. Equivalent plots of CiFi and Pore-C can be found in Figure S9. (C) UCSC Genome Browser view of 1q21.1-q22 (chr1:142.3 Mbp-150.4 Mbp; T2T-CHM13v2) with TopDom¹⁶ topologically associating domain (TAD) classifications (“domain” in black, “boundary” in green, “gap” in tan), centromere satellites (cenSat), and segmental duplications (SDs) track depicted. Two genomic “hotspots” are delineated, including the disease-associated TAR syndrome locus (purple) and chromosome 1q21.1 deletion (del) and duplication (dup) region (blue). TADs are also depicted as yellow triangles along the contact matrix. Pairwise interactions for CiFi are depicted on top and Hi-C on the bottom at 10-kbp resolution.

To better understand the extent to which CiFi improves assessment of chromatin interactions, we cataloged genomic regions classified as “gaps” in the TAD analysis—representing loci with insufficient data to call domains. Gaps are found less frequently in TADs delineated using CiFi (~2% of human autosome basepairs) compared with Illumina Hi-C (~5%), with more marked differences across repetitive regions (SDs: 8.6% CiFi vs 18% Hi-C; centromeres: 15% CiFi vs 34% Hi-C). These gaps present challenges in analysis of human genomic hotspots—regions enriched for SDs that

carry some of the greatest amounts of structural polymorphism in the genome ²⁰—where we observed none across flanking SDs using CiFi, in contrast to 8% using Hi-C.

One such locus is present on chromosome 1q21.1 where recurrent copy-number variants are associated with human developmental conditions ^{21,22}, including intellectual disability, autism, congenital abnormalities, and thrombocytopenia absent radius (TAR) syndrome (Figure 3C). While Hi-C TAD analysis of this locus contains mostly gaps due to insufficient interaction data, CiFi TAD analysis shows no gaps. Even in regions where Hi-C TAD domains are identified, discrepancies exist, e.g. across the ~200 kbp TAR syndrome locus, Hi-C classifies the unique portion of the locus as a domain flanked by two gaps spanning the adjacent SDs mediating the disease-associated microdeletion. In contrast, CiFi assigns the locus as a TAD “boundary”, a region that serves as a barrier between adjacent TADs. While further work is necessary to validate this finding, boundaries have been shown to be enriched for active chromatin and highly expressed genes ^{23,24}. Several housekeeping genes reside at this locus (*POLR3GL*, *POLR3C*, *RBM8A*), thereby supporting the CiFi result. The use of CiFi to characterize chromatin interactions at genomic hotspots such as this opens exciting new directions for understanding how structural variants impact gene regulation and underlying disease etiologies.

Additionally, we have added completely new analyses demonstrating that the multi-contact interactions produced by CiFi dramatically improves the ability to haplotype phase compared with traditional pairwise interactions produced by Hi-C by as much as 8-fold. We include this in the **Results** (p. 5, line 13):

To assess if the longer reads produced by CiFi improve haplotype phasing, we used a GM12878 telomere-to-telomere assembly ¹⁴ to assign phase to the mapped reads ¹⁵; this resulted in 23.9% CiFi phased segments versus 6.9% of phased Illumina Hi-C reads. Previous work has shown that ~98% of Hi-C read pairs mapping to homologous chromosomes represent *cis* interactions, with this proportion increasing to over 99.4% for pairs within 30 Mbp ⁵. This high rate of *cis* interactions enables haplotype assignment by linking phased reads with their unphased paired reads. We applied a similar principle to CiFi data and found that 97.8% of reads mapping to the same chromosome share the same haplotype when comparing phased segments within individual CiFi reads. Building on this observation, we imputed phase information to homologous segments within CiFi reads that mapped within 30 Mbp of segments with concordantly assigned phases (see Methods). This approach enables us to conservatively assign haplotype phase to 80.3% of CiFi segments compared to 10.9% of Hi-C reads (in line with published results for the Illumina dataset ¹³). Cumulatively, improved phasing has implications in characterizing chromatin structure using diploid genomes and producing high-quality phased genome assemblies.

I typically disclose my identity to authors: A. Papantonis

Reviewer #3 (Remarks to the Author):

CiFi is a protocol that provides a real improvement in the efficiency of long-read multi-contact Hi-C and reduces the input requirements, making it possible to sequence single insects. I am supportive of publishing this paper in Nature Communications as long as the following issues are addressed.

Critiques:

1. Figure S5: There seems to be a small population of regions that has more contacts in DpnII Hi-C than in DpnII CiFi (Figure S5), above the diagonal. What is the nature of these regions? They are not made up of mostly centromere sequence as the ones enriched in CiFi.

Thank you for noting this subset of discordant contact numbers not explained by SDs or centromeric sequences. By highlighting windows by chromosomes, we noticed that a majority of these spurious datapoints exhibiting increased coverage in Hi-C vs CiFi derived primarily from three chromosomes: 17, 19, and 22 (Figure R1A). Upon further investigation, we found that these chromosomes exhibit higher coverage in the Hi-C datasets relative to other chromosomes not observed for CiFi or Pore-C (Figure R1B). While the reason for this pattern is unclear, we note that these chromosomes exhibit the highest gene density and GC-content in the human genome, which may contribute to these results. Although additional analysis of other Hi-C datasets is required to verify if this is a common phenomenon, we do not believe it to be a technical artefact, as we obtained consistent results using two different read-mapping approaches (minimap2 and BWA-mem). Further, we note that following read-depth-based normalization, overall chromosomewide coverage differences would not impact smaller-scale interaction analyses such as TAD discovery.

2. Methods: The versions of packages used should be given in addition to the package names—this is inconsistent throughout the methods. The sequence data processing section should be more detailed, especially in terms of the aligner parameters.

Versions of packages have now been added to the methods. We have also expanded the sequence data processing to include additional details as well as aligner parameters.

3. The summary parameters are not enough to fully evaluate how well the libraries from low numbers of mammalian cells capture genome features. Please provide balanced contact matrices for the scaled-down GM12878 experiments.

We have added KR normalized contact matrices for the scaled-down GM12878 experiments across chromosome 1. These matrices show that overall contact signal and topologically associating domain (TAD) architecture is conserved as the number of cells is reduced (Figure S11).

Figure R1. Correlation of chromatin contacts between DpnII CiFi and Hi-C. (A) Analyzing the R^2 across all windows at 2.5 Mbp resolution highlights a subset of windows on human chromosomes 17, 19, and 22 showing discordant contact numbers between Hi-C and CiFi. **(B)** Normalized coverage per chromosome shows increased Hi-C reads mapping to the three chromosomes compared with CiFi and Pore-C.

Figure S11. Comparisons of chromatin contacts for human LCL GM12878 across different cell titrations for chromosome 1. Contacts are normalized using Knights-Ruiz algorithm. The starting cell amounts are listed on the left ranging from 62K to 5M. The 5M original CiFi protocol matches what was done for the initially successful CiFi run.

Minor/Technical issues:

4. Page 3, Line 17: Isn't a pairwise interaction represented by a read pair? Therefore 136 pairwise interactions should correspond to 136 Illumina *read pairs* and 272 *individual Illumina paired end reads*.

Yes, we agree and have updated the text to say “paired-end reads” as opposed to read pairs.

5. Figure 2B: Why is the unscaled trans-chromosomal contact signal weaker in CiFi than in Hi-C?

In our original pairwise interaction comparisons, we observed increased intrachromosomal relative to total genome-wide interactions for CiFi (78%) versus Hi-C (71%), which likely contributes to the differing trans-chromosomal contact signals in Figure 2B (now **Figure 3B**). To determine whether this was a characteristic of CiFi library preparation, we performed a similar comparison using the *Anopheles coluzzii* pairwise interactions and found the opposite pattern: the proportion of intrachromosomal interactions was lower for CiFi (66.5%) than Hi-C (90.6%). This difference is evident in the increased CiFi trans-chromosomal contact signals shown in **Figure 4C**. Since the inter- to intra-chromosomal ratio serves as a standard quality control metric for 3C libraries, these differences likely reflect technical variations of individual experiments. Further investigation of this phenomenon will be important as additional CiFi datasets become available.

6. Figure S6: The triangles are gray, not yellow as described in the caption.

Thank you for finding this discrepancy. We have updated the text to say “light-blue” triangles instead of yellow.

7. How much DNA was obtained from each half of the fly? The amount of DNA is given for the mosquito but not the fly.

These details of DNA yields for the HiFi and CiFi tissue inputs have now been added to the Results (p. 8, line 18):

In splitting the sample, half of the tissue yielded 1.35 μg of high-molecular-weight DNA that was used for SMRTbell library preparation for HiFi whole-genome sequencing data generation. The post-crosslink reversal purified DNA from the other half of the tissue was quantified at 1.11 μg and used for CiFi library generation.

8. It is not entirely clear how the library columns in Table S3 correspond to the ones in Table S1.

To improve interpretation of information, we have added corresponding “Library ID” rows in Tables S1 and S3. In some cases, data was combined across multiple libraries, which is indicated in Table S3.

Reviewer #1 (Remarks to the Author):

The revised manuscript detailing the CiFi protocol and its applications in investigating genome topology and scaffolding and phasing of de-novo genome assemblies is in a much better shape. I really appreciate the extra details in the mapping quality and coverage of all technologies across the different regions of the genomes presented, as well as the more in-depth analysis of recovery and discover of TADS with the CiFi protocol compared to libraries from other technologies.

We thank all reviewers for their positive comments and insightful feedback that ultimately improved the manuscript and resource overall.

I am happy now that the claims made by the authors are well justified by the data presented.

We are appreciative of Reviewer 1's final comments.

I only have a couple of minor points:

Page 9 line 2: The quoted number of contacts should be 70,000, not 30,000. The value of 70k is stated in the rebuttal, but not the main text. One could also add "... compared to 300-500k Hi-C interactions", but I'll leave this to the authors.

Thank you for pointing this out. We have made the correction to 70K instead of 30K and added the comparison to Hi-C interactions.

Results (p. 6, line 36)

Notably, subsampling to as low as 70,000 proximity-ligated pairs still achieved a near chromosome-scale scaffolded assembly compared to ~300,000 Hi-C interactions (Supplementary Figure 14)

Please add the <https://github.com/sheinasim-USDA/CiFi2PE> repository to the "Code Availability" section to improve findability.

We added the link to the CiFi2PE repository to the "Code Availability" section and a corresponding Zenodo link.

CODE AVAILABILITY (p. 14, line 5)

Bioinformatic pipelines are available <https://github.com/mydennislab/CiFi/tree/main> and <https://github.com/sheinasim-USDA/CiFi2PE> and have been deposited at Zenodo (DOI <https://zenodo.org/records/17526543>⁵⁹).

Reviewer #1 (Remarks on code availability): The tools and arguments are well detailed and will be useful to the community in analysing similar data or attempting to reproduce the steps in this paper.

Reviewer #2 (Remarks to the Author):

I find that the authors have done a commendable job in addressing critiques by all reviewers, and I fully support publication of their revised manuscript in Nat Commun.

We thank Reviewer 2 for their positive feedback and support.